# communications
# earth & environment

# Stable ocean redox during the main phase of the Great Ordovician Biodiversification Event

Álvaro del Rey [1], Christian Mac Ørum Rasmussen [1], Mikael Calner [2], Rongchang Wu[3], Dan Asael[4] & Tais W. Dahl [1✉]

The Great Ordovician Biodiversification Event (GOBE) represents the greatest increase in marine animal biodiversity ever recorded. What caused this transformation is heavily debated. One hypothesis states that rising atmospheric oxygen levels drove the biodiversification based on the premise that animals require oxygen for their metabolism. Here, we present uranium isotope data from a Middle Ordovician marine carbonate succession that shows the steepest rise in generic richness occurred with global marine redox stability. Ocean oxygenation ensued later and could not have driven the biodiversification. Stable marine anoxic zones prevailed during the maximum increase in biodiversity (Dapingian–early Darriwilian) when the life expectancy of evolving genera greatly increased. Subsequently, unstable ocean redox conditions occurred together with a marine carbon cycle disturbance and a decrease in relative diversification rates. Therefore, we propose that oceanic redox stability was a factor in facilitating the establishment of more resilient ecosystems allowing marine animal life to radiate.

[1] GLOBE Institute, University of Copenhagen, Øster Voldgade 5-7, DK-1350 Copenhagen K, Denmark. [2] Department of Geology, Lund University, Sölvegatan 12, SE-223 62 Lund, Sweden. [3] Nanjing Institute of Geology and Palaeontology, Chinese Academy of Sciences, 39 East Beijing Road, Nanjing 210008, China. [4] Department of Earth and Planetary Sciences, Yale University, New Haven, CT 06511, USA. ✉email: tais.dahl@sund.ku.dk

The Great Ordovician Biodiversification Event was the greatest accumulation of marine metazoan richness in Earth's history[1]. Two widely different views on the onset and duration of the GOBE has been proposed: A slow, continues rise in richness over >40 million years through the Cambrian–Ordovician periods is observed based on broadly binned temporal analyses of early Paleozoic fossil occurrences[2,3]. Opposing this view stands studies conducted with high temporal resolution that show a rapid radiation that started during the Middle Ordovician[1,4,5]. Well-resolved richness datasets notably show two interesting patterns: firstly, a prelude to the GOBE occurred in South China and in the adjacent peri-Gondwanan areas[1,6,7], but generic richness did not rise markedly until well into the early Ordovician[7,8]. Secondly, global metazoan richness data constrain the main onset and bulk of the GOBE outside South China and peri-Gondwana to have occurred within maybe as little as two million years during the earliest part of the Middle Ordovician Darriwilian Age[9–10]. It is this key interval in Earth history which is the focus of the current study.

Generic richness increased four-fold during the Middle–early Late Ordovician[11,12] and there was a complete restructuring of trophic relationships[1–3]. The first complex marine ecosystems with deeper tiering, intricate trophic cascades and ecospace partitioning were established[13]. Whereas the Cambrian–early Ordovician faunas had been dominated by detritus-feeding benthos, from the mid-Ordovician onwards predators and suspension-feeding organisms vastly radiated not just on the seafloor but also throughout the water column[14]. Reef systems started resembling modern analogs, transitioning from algal and bacterial mats to ones with complex frameworks constructed by metazoans[15]. What instigated this transformative ecological expansion in the marine environment is therefore a fundamental question to answer. A number of different causal mechanisms have been proposed[1], including two abiotic factors that have gained momentum. One, climatic cooling inducing a decrease in tropical sea surface temperatures from Cambrian and Early Ordovician hot conditions (~42 °C) to temperatures similar to present-day (27–32 °C) during the Middle Ordovician Darriwilian Age, is thought to have provided more favorable conditions for marine animal life thus promoting the radiation[16–20] (Fig. 1), for example via temperature-dependent oxygen supply and metabolic demand[21,22]. And two, an increase in atmospheric oxygenation during the Middle Ordovician, in contrast to

concurrent anoxic events and elevated extinction rates from the late Cambrian to early Ordovician[23], has been proposed as a driver of the GOBE[24] (Fig. 1). This is justified either by higher molecular oxygen ($O_2$) levels in surface waters fueling a more energetic lifestyle (e.g. carnivory) and allowing higher trophic levels, and/or by oxygenating new seafloor areas, creating new ecospace for marine animals in oceans with larger oxygen-depleted zones than today[25–27]. Because animals depend on $O_2$ to survive[28,29] and marine $O_2$ levels generally decline with ocean depth as $O_2$ is merely consumed below the photic zone, a more oxygenated world during the Middle Ordovician might have supported more animal life on the continental shelves. Nevertheless, a recent paleontological study has suggested that there is a time difference between the onset of the GOBE and the rise in atmospheric $O_2$[10] and thus, the role of $O_2$ as a direct causal mechanism in the rise of biodiversity remains unclear.

Here we assess the relationship between oceanic oxygenation and animal radiation during a ~10 million year-long time interval during the main phase of the GOBE[1,9,10] by analyzing uranium (U) isotopes in 55 marine carbonate beds from a sedimentary succession deposited in the Middle Ordovician (Dapingian–Darriwilian). The U isotope composition of marine carbonates has been documented to respond to global changes in the oceanic redox landscape (e.g.[30–32]). By comparing the U isotope composition of marine carbonates from the Middle Ordovician with high-resolution global generic richness data, we examine whether or not changes in the global oxygenation state of the oceans were related to the expansion of marine animal life of the GOBE.

**Uranium isotopes in marine carbonates as a proxy for ocean oxygenation.** The U isotope signature of marine carbonates ($\delta^{238}U_{CARB}$) can inform us about the global oxygenation state of the oceans because the marine U removal in anoxic and sulfidic (euxinic) settings induces a large positive U isotope offset in sediments[33], leaving behind U in seawater with an isotopically distinct signature. In general terms, lower $\delta^{238}U_{CARB}$ values than modern carbonate sediments ($\delta^{238}U_{CARB} = -0.14 \pm 0.11‰$, 1sd)[34] indicates deoxygenation or an expansion of anoxic water masses, whereas higher $\delta^{238}U_{CARB}$ values similar to modern-day carbonates implies a lesser extent of euxinia or greater proportion of sediment burial in the oxygenated parts of the oceans[35]. Microbial U reduction involves large isotope fractionation[36,37]. Because U(VI) is predominantly soluble and U(IV) insoluble in

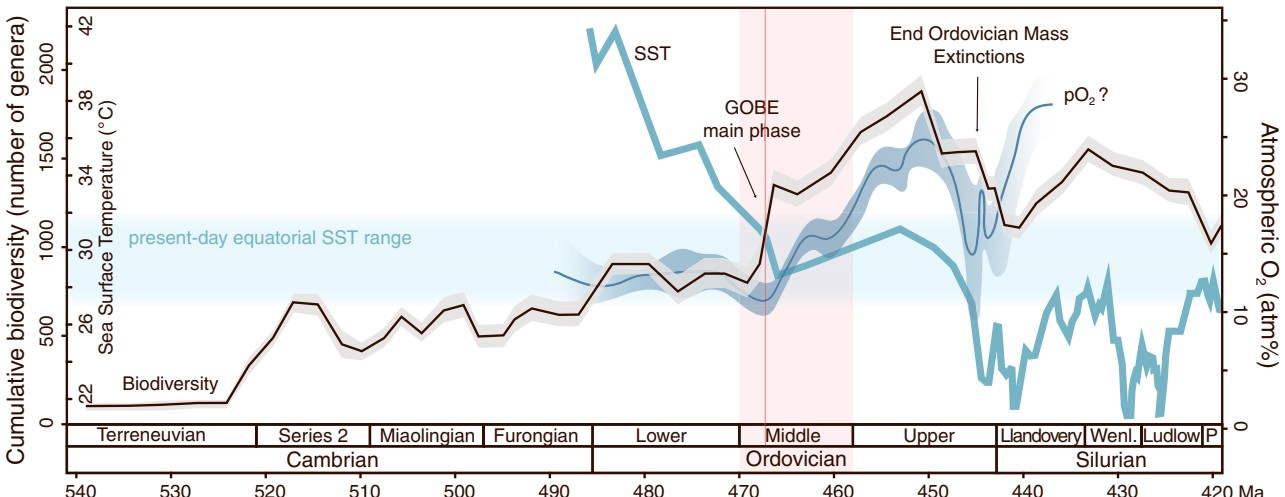

**Fig. 1 Biodiversity accumulation and environmental change during GOBE.** Cumulative biodiversity[10], sea surface temperatures[16,67] and inferred atmospheric oxygen levels[24] throughout the early Paleozoic (modified from ref. [21]). The pink area highlights the Middle Ordovician. The red vertical line represents the Dapingian-Darriwilian boundary. P Pridoli, SST Sea Surface Temperature.

the oceans, the reduction process takes U out of solution and simultaneously fractionates its isotopes with the heavier $^{238}$U isotope enriched in the product[36,38]. Therefore, $^{238}$U is preferentially removed in euxinic settings where U accumulates into the sediments, leaving behind $^{238}$U-depleted ($^{235}$U-enriched) waters. Hence, sediment burial under reducing conditions becomes the main sink of U associated with isotope fractionation[33,39]. The incorporation of U into biotic or abiotic carbonates, although also a large U sink[40], only involves limited isotope fractionation[34,41,42]. Thus, the extent of euxinic water masses becomes the main control of the U isotope composition of the oceans[38]. Lastly, due to its much longer residence time (~400 kyr) compared to ocean mixing time (~1.5 kyr), U behaves conservatively and has a globally uniform isotopic distribution in the ocean[38]. Therefore, marine carbonates formed under oxic conditions record the isotope composition of seawater at the time of deposition, which despite acquiring a documented range of isotopic offsets during early carbonate diagenesis[34,43], it represents a measure of the global oxygenation state of the oceans[35].

**Geological setting**. The analyzed carbonate rocks from the Middle Ordovician consist of a continuous sedimentary succession from the Kinnekulle table mountain in southern Sweden (Kinnekulle-1 drillcore, Fig. 2a) that represent a condensed cool-water carbonate facies deposited with limited terrigenous input within the Baltoscandian Paleobasin[44,45] (Fig. 2b). The carbonate succession includes numerous mineralized hardgrounds and comprises a fairly monotonous succession of mudstones and wackestones with subordinate marly limestones and calcareous shales. The basal part of the studied section (Fig. 3) consists of the informal, topostratigraphic unit Lanna limestone (Dapingian–lower Darriwilian; Dw1 stage slice sensu ref. [46]) overlain by the Holen limestone (Dw1–upper Dw2). The Lanna–Holen interval is characterized by a reddish-brown color and a rather homogeneous macroscopic appearance. Within the lower Holen limestone, a conspicuous and discrete gray limestone interval is referred to as the Täljsten[44]. The gray-colored Skövde limestone (middle Dw3) disconformably overlies the Holen limestone, evidencing a hiatus at Kinnekulle encompassing the lower Dw3[44]. The top of the studied section comprises the gray basal marly/nodular limestone of the Gullhögen Formation (upper Dw3) deposited conformably over the Skövde limestone.

The depositional environment for the Lanna and Holen limestones was an open marine, fairly deep setting with low depositional rates (3–5 mm/kyr)[45,47] and recurrent episodes of non-deposition and formation of corrosional hardgrounds. The erosional surface and associated hiatus at the top of the Holen limestone attests to a longer period of erosion and non-deposition. The alternating red or gray sediment color may have been caused by sea-level change through changing accumulation rates and early diagenetic redox potential[48].

## Results

**A δ$^{238}$U record through the Middle Ordovician**. The δ$^{238}$U curve from the carbonate succession of Kinnekulle (Fig. 3; Table S1) exhibits secular trends that can be subdivided into two distinct intervals depending on the amplitude or stability of their δ$^{238}$U variations (Fig. 3). In the lower part of the section up until the Täljsten (Interval I, Dp1–Dw1), δ$^{238}$U trends are steady with a lower average value and smaller variance at $-0.36 \pm 0.09$‰ (1sd, $n = 26$) than observed in modern carbonates from the Bahamas ($-0.14 \pm 0.13$‰, $n = 150$; Table S2). The smaller variance in Interval I compared to today's carbonate sediments is statistically significant ($p < 0.05$, F-test). The following interval (Interval II, Dw2–Dw3) shows unstable δ$^{238}$U values with large amplitude fluctuations (from $-0.78$ to $0.16$‰). Interval II demonstrates a larger variance ($p < 10^{-4}$, F-test) than modern carbonates that have experienced burial diagenesis, but not meteoric diagenesis (Higgins et al. 2018; Table S2). The systematic δ$^{238}$U trends in Interval II could therefore reflect secular changes in ancient seawater composition. There is a positive trend during the Dw2, followed by a negative one continuing to the unconformity at the Dw2–Dw3 boundary. Finally, a large positive δ$^{238}$U trend overlain by considerable fluctuations occurs throughout Dw3 indicative of a change towards globally more oxygenated ocean conditions.

Elemental data shows elevated U content in the gray limestone facies of the Täljsten and Skövde limestone–Gullhögen Formation associated with higher δ$^{238}$U variability. The secular δ$^{238}$U changes are not coupled to variations in elemental ratios (Mg/Ca, Sr/Ca, Mn/Sr, Al/Ca, Mo/Ca) (Supplementary Fig. 1) or facies change, indicating that the δ$^{238}$U trends are not systematically altered by changes in primary carbonate mineralogy, chemical alteration by diagenetic fluids, terrigenous input, or local redox conditions (see Supplementary Information for all isotopic and geochemical data and further discussion about diagenesis).

## Discussion

**The global oxygenation state of the oceans and changes in marine biodiversity**. A lower δ$^{238}$U value of Mid-Ordovician carbonates relative to modern carbonates points to a generally more oxygen-depleted ocean than today. With a greater proportion of global oceanic U burial in anoxic and sulfidic (euxinic)

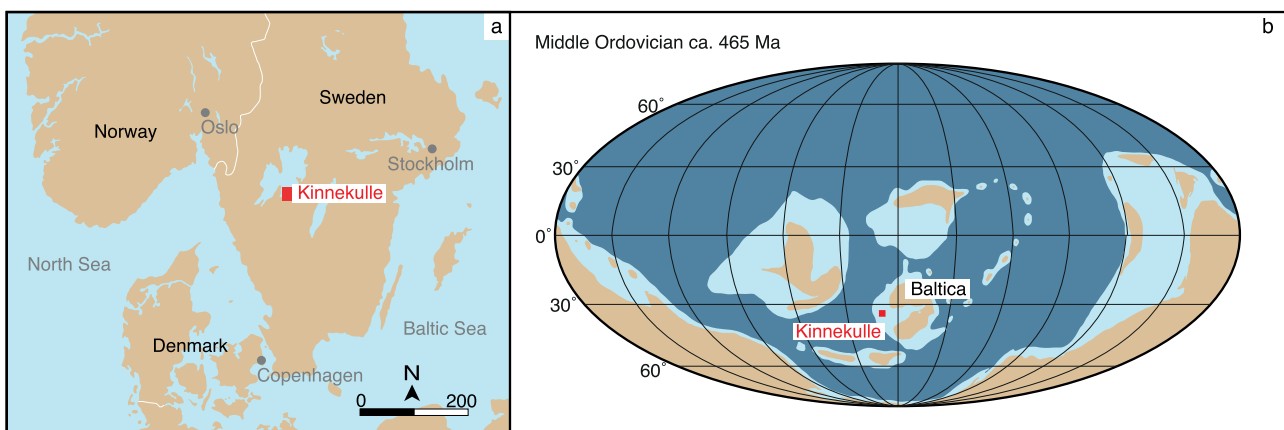

**Fig. 2 Study area. a** Location of Kinnekulle, province of Västergötland, Sweden. **b** Paleocontinental reconstruction of the Middle Ordovician (ca. 465 Ma)[68]. The continent of Baltica is indicated next to the approximate location of the Baltoscandian paleobasin. The red area indicates Kinnekulle.

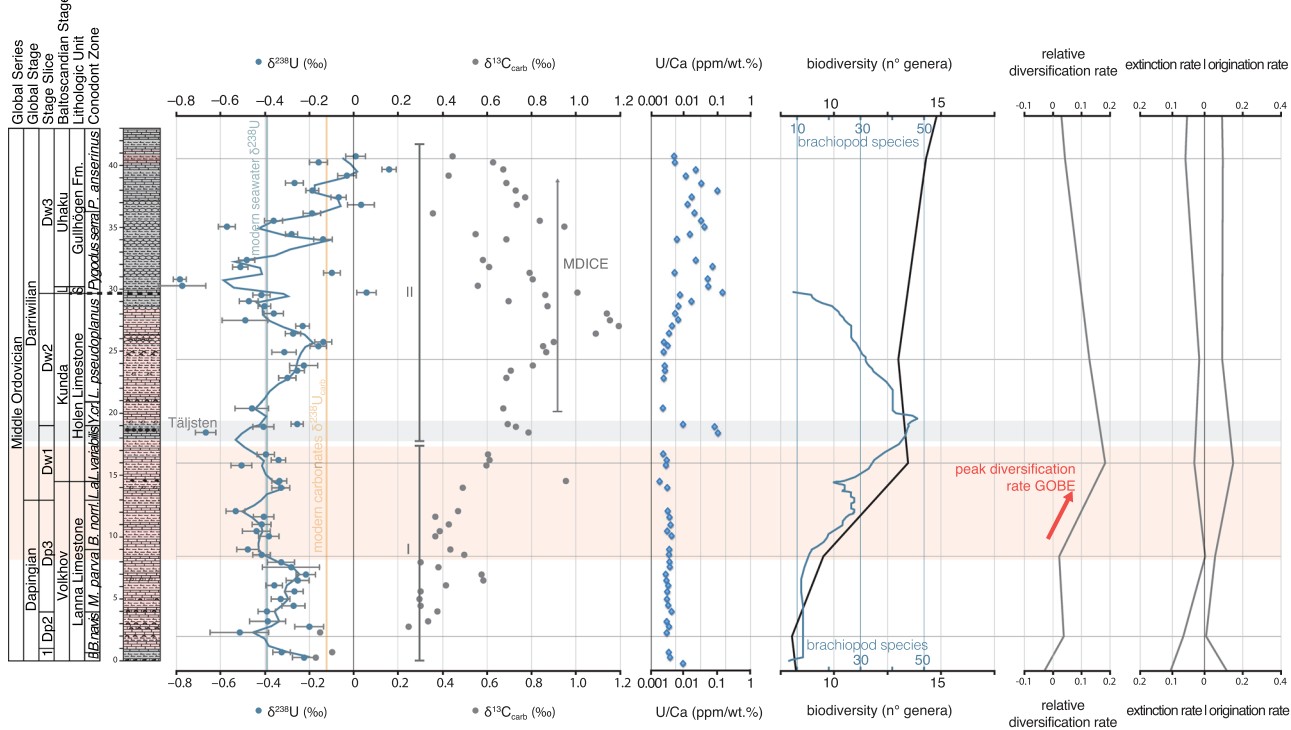

**Fig. 3 Geochemical results compared to published biodiversity data.** Stratigraphy and geochemical data ($\delta^{238}U$, $\delta^{13}C$, U/Ca) across the Kinnekulle-1 drillcore (N 58°35.97792', E 13°25.00083'). Conodont biostratigraphy after ref. [56]. and references therein. Stage-level marine animal biodiversity (black, gray curves) and brachiopod richness (blue curve) after refs. [9,10,57]. Horizontal lines represent the middle point of time bins to which biodiversity curves were calculated[9,10]. Stage slices sensu ref. [46]. Modern seawater $\delta^{238}U$ ~ −0.39‰[40], modern carbonates $\delta^{238}U_{CARB}$ ~ −0.14‰[34]. Intervals I and II are shown next to the $\delta^{238}U$ data. The dashed black line in the stratigraphic column marks the unconformity of the section. The blue curve corresponds to the spline interpolation of the $\delta^{238}U$ data points. Error bars for $\delta^{238}U$ represent analytical precision (2 s.e.). NB! The U/Ca data is shown on a log-scale to emphasize the details. Dp: Dapingian, Dw: Darriwilian, 1: Dp1, L: Lasnamägi, S: Skövde Limestone, *B.: B. triangularis, B. norrl.: Baltoniodus norrlandicus, L.a.: L. antivariabilis, Y. cr.: Y. crassus, P. anserinus.: Pygodus anserinus.*

settings, this situation is not too dissimilar from that recorded in the earliest Ordovician[49] and the Silurian[50–52]. The Dapingian and Darriwilian $\delta^{238}U$ record shows a significant shift from an interval with a relatively stable global ocean oxygenation state (particularly from middle Dp3–upper Dw1) to unstable redox conditions characterized by large fluctuations in $\delta^{238}U$ values (Dw2–Dw3) (Fig. 3). A continuous drift towards more oxygenated oceans follows in the late Darriwilian (Dw3). Thus, an increase in oxygenation could not have triggered the earliest Darriwilian inception phase of the GOBE as previously suggested[24]. Both global and regional biotic richness data shows that the peak of this main phase occurred during the late Dw1[1,9,10,53]. Therefore, we find that a relatively stable global oxygenation state of the oceans prevailed during the period leading up to the maximum increase of biodiversity of the GOBE (Interval I, Fig. 3).

After this stable $\delta^{238}U$ phase, biodiversity and relative diversification and origination rates decrease when large $\delta^{238}U$ fluctuations are observed (Interval II, Fig. 3). This change coincides with the onset of a perturbation of the global marine carbon cycle known as the "Middle Darriwilian Isotope Carbon Excursion" or MDICE (from approximately in the early Dw2)[54–56]. Thus, relative diversification slows down when the redox landscape becomes unstable, and the carbon cycle is globally disturbed. Further, extinction rates increase, and brachiopod richness[57,58] reaches a low when the $\delta^{238}U$ curve suggests a shift towards more euxinic water masses in the oceans, around the unconformity of the section (Dw2–Dw3 boundary) (Fig. 3).

The transition from a relatively stable to unstable redox landscape and accompanying changes in marine biodiversity

occurs in the conspicuous Täljsten interval within the lower Holen limestone near the Dw1–Dw2 boundary (Fig. 3). The Täljsten possess a series of atypical facies and paleontological features that evidence a period of stressed environments, including opportunistic disaster taxa, such as the echinoderm genus *Eosphaeronites* occurring in rock forming quantities, suggesting the collapse of marine ecosystems[44,59]. We see this expressed in a rapid and large $\delta^{238}U$ fluctuation within these beds and elevated abundances of redox sensitive elements (U/Ca, Mo/Ca, P/Ca, Supplementary Fig. 2), which suggest reducing conditions in the local depositional environment[59]. In addition, a similar lithological change is also observed in coeval limestones in South China[58,60]. Therefore, the abrupt lithological shift in the rather uniform carbonate succession in two distinct and widely separated paleocontinents suggests the transition from stable to unstable conditions was initiated by a global event that disrupted sedimentary cycles and the marine redox landscape.

**Animal life thrived during the Dapingian–early Darriwilian.** An important factor that has been causally related with the onset of the GOBE is climatic cooling reaching tropical sea surface temperatures similar to present day in the Middle Ordovician, providing more favorable conditions for marine animal life[16,20,61,62]. Modern-like equatorial sea surface temperatures were attained from around the late Dapingian[16] and a 4–5 °C cooling affecting the Baltoscandian Paleobasin was identified in strata just above the Dapingian–Darriwilian boundary[20,61]. Thus, ocean cooling occurred in the same time interval with the comparatively most stable ocean redox conditions and the maximum increase in marine animal genera (middle Dp3–upper Dw1).

During the Dapingian–early Darriwilian, the life expectancies of genera reached an early Paleozoic maximum just prior to the onset of the GOBE[9]. Thus, generic lineages survived longer at the time with the most stable $\delta^{238}U$ trends (Fig. 3). This meant that marine ecosystems became more resilient, favouring greater biodiversity[9]. Therefore, we suggest ecosystems evolved and became more resilient, persisting for longer time spans, because animals thrived in the more stable marine redox landscape and under more clement tropical ocean temperatures, providing a higher metabolic scope for animals in greater parts of the oceans[20–22] (Fig. 3).

## Conclusions

The comparison between the $\delta^{238}U$ of marine carbonates and marine biodiversity accumulation through the Middle Ordovician shows that the inception and main phase of the GOBE was not linked to an increase in ocean oxygenation, but to a rather stable marine redox landscape. An oxygenation event in the oceans and atmosphere[24] is recorded after and therefore, it could not have driven the biodiversification event. The GOBE is also marked by a notable peak in the life expectancies of genera just before the onset of the radiation and in the time interval with the most stable $\delta^{238}U$ trends (Dapingian–early Darriwilian), prior to a global disturbance of the marine carbon cycle (pre- MDICE) and when cooling oceans reached modern-like equatorial sea surface temperatures. Therefore, we suggest that the relative stability of the marine redox landscape over >1 Myr was one of the fundamental factors allowing more time for complex ecological communities to evolve by preventing the expansion of anoxia on shorter timescales. Extinction pulses occurred frequently during much of the Cambrian–Early Ordovician interval[9], which is in sharp contrast to the Middle Ordovician where extinction levels appear to be at a minimum. Thus, a stable marine redox landscape in more temperate oceans was likely key in promoting the greatest marine animal radiation in Earth's history.

## Methods

Analyzed samples for U isotopes correspond to ca. 1 g powdered carbonate rock each. Powders were extracted from the rocks' matrices wherever possible, avoiding large grains and areas that macroscopically evidence secondary alteration (e.g., weathering, veins, etc.). Samples were dissolved using a soft leaching procedure using ca. 45 mL 0.5 M HCl to avoid the dissolution of any non-carbonate fraction, but achieving complete dissolution of the carbonates[35]. A small aliquot (0.15%) of the leachate was taken for major and trace element concentrations analyses and the rest (99.85%) was used for uranium isotope analyses.

Major and trace element concentrations were analyzed using high-resolution sector field inductively coupled plasma mass spectrometry (ICP-MS), Thermo-Fisher Scientific Element XR, at the Yale Metal Geochemistry Center, Yale University. Measurement precision was generally better than 5% (2σ). BHVO-2 and NOD-A-1 USGS geostandards were processed along with samples in each run using a standard sample introduction system[63] and were within ±5% of reported values.

For U isotope analyses, U was extracted and purified from the carbonate leachates using the UTEVA ion exchange resin procedure and measured using the $^{233}U$-$^{236}U$ double-spike method[64–66] on a ThermoFisher Scientific Neptune Plus Multicollector ICP-MS at the Yale Metal Geochemistry Center, Yale University. Up to 150 ng U per sample were purified passing through UTEVA ion exchange resin twice after adding 240 μL double-spike solution per 150 ng U to have the same sample/spike ratio in all samples and standards. The IRMM-3636 $^{233}U$-$^{236}U$ double-spike method is used to correct for instrumental mass bias and potential isotopic fractionation during column chemistry[64]. In summary, after adding the spike, samples were dissolved in 7 M $HNO_3$ and then loaded into 1 mL UTEVA-resin-loaded columns pre-cleaned with 8 mL 0.05 M HCl and pre-conditioned with 7 M $HNO_3$. Matrix elements (except thorium) were eluted using 15 mL 3 M $HNO_3$ and Th using 6 mL 5 M HCl. U was then released from the resin using 13 mL 0.05 HCl. The final solutions with collected U were then dried down and treated with small amounts of $H_2O_2$ and concentrated $HNO_3$ to eliminate potential residual organic compounds leached from the resin. This procedure was applied twice and after the second time, samples were dried down and brought up in 3 mL 5% $HNO_3$ (50 ppb U) for isotope composition analyses. Samples were introduced to the multicollector ICP-MS using an ApexIR sample introduction system and measured at low resolution. Isotopes $^{232}Th$, $^{233}U$, $^{235}U$, $^{236}U$ and $^{238}U$ were measured

simultaneously on Faraday collectors connected to $10^{11}$ Ω amplifiers achieving a signal of ~25–45 volts on $^{238}U$, depending on concentration. Procedural blanks were 10-40 pg U. The data are reported as parts per thousand (‰) deviation of sample's $^{238}U$/$^{235}U$ ratio relative to the CRM-145 reference standard ($\delta^{238}U$). Accuracy and external reproducibility of the data were evaluated by analyzing Ricca ($-0.23 \pm 0.08$‰, 2sd long-term reproducibility, n = 12) and CRM-129a ($-1.74 \pm 0.12$‰, 2sd long-term reproducibility, $n = 15$) reference materials processed through the same chemical purification procedure with the same sample/spike ratio as the studied samples.

## Data availability

All data generated or analysed during this study are included in this published article (and its supplementary information files).

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

## Acknowledgements

We thank Xuefang Wu for laboratory assistance at Yale University during the COVID-19 pandemic. We also thank Anders Lindskog for his work and discussions on the conodont biostratigraphy. AdR and TWD acknowledge financial support from the Carlsberg Foundation through its Distinguished Associate Professor program to TWD (CF16–0876). TWD thanks the Danish Council for Independent Research (7014-00295B and 8102-00005B). CMØR is grateful for funding from GeoCenter Denmark (2015-5 and 3-2017). This is a contribution to the IGCP Project 735 'Rocks and the Rise of Ordovician life (Rocks n' ROL)'. MC and RW acknowledge the Crafoord foundation (20140806) and the Royal Physiographic Society of Lund, respectively, for funding the recovery of the Kinnekulle-1 drillcore. The Department of Geology at Lund University made the drill-core available for sampling.

## Author contributions

This study was designed by T.W.D. with input from M.C. and C.M.Ø.R. Sampling and isotope analyses preparation done by A.d.R. Isotope measurements made by D.A. Conodont biostratigraphy made by R.W. A.d.R. and T.W.D. wrote the paper with input from all co-authors.

## Competing interests

The authors declare no competing interests.
