## [Peer Review File · Communications Earth & Environment]

Web links to the author's journal account have been redacted from the decision letters as indicated to maintain confidentiality.

14th Mar 22

Dear Dr Dahl,

Please allow me to apologise for the long delay in sending a decision on your manuscript titled "Stable ocean redox during the main phase of the Great Ordovician Biodiversification Event". It has now been seen by 3 reviewers, and I include their comments at the end of this message. They find your work of some potential interest, but important points are raised. We are interested in the possibility of publishing your study in Communications Earth & Environment, but would like to consider your responses to these concerns and assess a revised manuscript before we make a final decision on publication.

We therefore invite you to revise and resubmit your manuscript, along with a point-by-point response that takes into account the points raised. Please highlight all changes in the manuscript text file.

In particular, please ensure that the revised manuscript meets the following editorial thresholds:

**** Provide a compelling case that the Great Ordovician Biodiversification Event occurred during stable environmental conditions rather than during a sharp rise in ocean oxygenation, including accounting for the potential for this result to be an artefact of low data density during the event ****

**** Fully justify your interpretations of the timing of the Great Ordovician Biodiversification Event in the light of relevant recent and previous work ****

**** Present robust evidence that your carbonate record is representative of an open ocean environment and therefore able to inform on global ocean redox conditions ****

Please use the following link to submit your revised manuscript, point-by-point response to the referees' comments (which should be in a separate document to any cover letter) and the completed checklist:

[link redacted]

We hope to receive your revised paper within six weeks; please let us know if you aren't able to submit it within this time so that we can discuss how best to proceed. If we don't hear from you, and the revision process takes significantly longer, we may close your file. In this event, we will still be happy to reconsider your paper at a later date, as long as nothing similar has been accepted for publication at Communications Earth & Environment or published elsewhere in the meantime.

We understand that due to the current global situation, the time required for revision may be longer

than usual. We would appreciate it if you could keep us informed about an estimated timescale for resubmission, to facilitate our planning. Of course, if you are unable to estimate, we are happy to accommodate necessary extensions nevertheless.

Please do not hesitate to contact me if you have any questions or would like to discuss these revisions further. We look forward to seeing the revised manuscript and thank you for the opportunity to review your work.

Best regards,

Joe Aslin

Senior Editor,
Communications Earth & Environment
<https://www.nature.com/commsenv/>
Twitter: @CommsEarth

EDITORIAL POLICIES AND FORMATTING

Editorial Policy: [Policy requirements](https://www.nature.com/documents/nr-editorial-policy-checklist.zip)

Furthermore, please align your manuscript with our format requirements, which are summarized on the following checklist:

[Communications Earth & Environment formatting checklist](https://www.nature.com/documents/commsj-phys-style-formatting-checklist-article.pdf)

and also in our style and formatting guide [Communications Earth & Environment formatting guide](https://www.nature.com/documents/commsj-phys-style-formatting-guide-accept.pdf) .

*** DATA: Communications Earth & Environment endorses the principles of the Enabling FAIR data project (<http://www.copdess.org/enabling-fair-data-project/>). We ask authors to make the data that support their conclusions available in permanent, publically accessible data repositories. (Please contact the editor if you are unable to make your data available).

All Communications Earth & Environment manuscripts must include a section titled "Data Availability" at the end of the Methods section or main text (if no Methods). More information on this policy, is available at <http://www.nature.com/authors/policies/data/data-availability-statements-data-citations.pdf>.

If a community resource is unavailable, data can be submitted to generalist repositories such as [figshare](https://figshare.com/) or [Dryad Digital Repository](http://datadryad.org/). Please provide a unique identifier for the data (for example a DOI or a permanent URL) in the data availability statement, if possible. If the repository does not provide identifiers, we encourage authors to supply the search terms that will return the data. For data that have been obtained from publically available sources, please provide a URL and the specific data product name in the data availability statement. Data with a DOI should be further cited in the methods reference section.

REVIEWER COMMENTS:

Reviewer #1 (Remarks to the Author):

The del Rey et al manuscript is a straightforward, well organized study testing the hypothesis that a global seawater oxygenation increase triggered the GOBE. The authors use $\delta^{238}\text{U}$ trends from Swedish marine carbonates and comparisons to previously reported biodiversity, origination/extinction data sets to address the question. The new $\delta^{238}\text{U}$ data spans a relatively short time interval in the Middle Ordovician (~12 My) and assumes that the main onset of GOBE biodiversity increase occurs near the Dapingian-Darriwilian boundary. The entire hypothesis of the study is premised on assuming the GOBE onset occurred during this short time interval. However, the results of a recent study by Servias et al (2021) suggest that this assumption requires re-evaluation because the various, widely used, biodiversity data sets are biased and/or incomplete. Instead, Servias et al argue that the Ordovician radiation was a complex, long-term process of multiple marine diversifications and that the idea of a relatively short, dramatic global radiation at a particular time interval is an oversimplification--neither was it short or coevally global, and the biodiversity data sets used in the past are biased and/or incomplete. Servias et al argue that depending on which biodiversity data set is used, the GOBE occurred anytime throughout the Early to Late Ordovician (Servias' et al figure 1). Given this, the key interpretation of this study of the timing of the GOBE onset is associated with stable seawater redox conditions rather than increasing $\delta^{238}\text{U}$ trends (indicating increasing oxygenation) is problematic. The authors need to clearly address

this issue, provide more explanation on the GOBE timing (and why they choose to use particular timing interpretations) and provide solid evidence to refute the Servias et al findings if they disagree.

Below are specific line-by-line comments.

Line 44—This is the first time the (assumed) GOBE timing is mentioned. A clearer discussion of the radiation and its timing is necessary as the entire study hinges on this.

Lines 66-69- The second potential driver (Middle Ordovician oxygenation) is provided then an argument on why it is not supported is given. This same pattern should be provided for the first hypothesis (SST cooling) and why is or is not supported.

Lines 73-81—this brief discussion of U isotopes as redox proxy is more appropriate in Background rather than in Introduction

76- be specific in stating a large fractionation occurs in euxinic sediments specifically, not general anoxic sediments.

77-78—not clear what modern values of -0.14 permil is referring to

93- euxinic specifically

102- 'known' is too optimistic, the 'most recently documented range of diagenetic offset' is better language

109- readers outside the carbonate community do not know what cool water carbonates means

119- need rock type information rather than just 'beds'

127- 'variations' cannot be stable, reword

128-129- Confusing sentence, average measured values may be similar to modern carbonates, but the variance is not similar. Needs clarification.

132-133- Best not to focus on the likely outliers within the Taljsten (for example similar point-by-point variability at ~3 m in Dp2). These large point-by-point variabilities at the sub-meter scale of >~0.2-0.3 permil are not realistic for the global oceans. Focus instead on the major and longer term excursions starting above Taljsten.

139- define gray facies

142—statement is too optimistic, best to say they are not significantly altered by....

154-157—see comments above about GOBE timing, this statement is based on the assumption that the main GOBE increase occurred in this relatively short time interval

159- Interval I is not labelled on Figure 3 or mentioned in figure caption.

161-169—entire discussion is based on assumption that diversity and extinction/origination rate trends of previous studies are correct and does not include Servias et al 2021 results

175—opportunistic disaster taxa needs explanation, what are they, what do they suggest and why?

181—to emphasis this, provide lateral distance between the two continents recording this event

186—when did GOBE really start?

192—see previous comments on GOBE timing

202-203—be specific on what SST define 'more favorable'. Cooler, warmer, transitioning?

208-209—see GOBE timing comments

216-221- run on sentence and confusing concepts, rewrite

Figures

1- Font size on all figures need enlargement

2- Figure 3. Pink band not labelled and Interval I mentioned in text is not shown

Maya Elrick

Reviewer #2 (Remarks to the Author):

Please find the full review report in the attached pdf file.

Reviewer #3 (Remarks to the Author):

This manuscript deals with the possible environmental forcings that triggered the Great Ordovician Biodiversification Event (GOBE). The authors quantify seawater redox by using a proxy method based on U isotopes in marine carbonates. They conclude that the maximum increase in biodiversity during the mid-Ordovician took place when marine anoxic zones were stable.

I cannot understand how biodiversity, the one based on metazoans needing oxygen for respiration, may increase in an anoxic context (line 28)! Meanwhile, $\delta^{13}\text{C}$ values of carbonates increase, reflecting an increase in marine productivity and long-term storage of organic matter. The authors refer to the paper of Edwards et al. (2017; Nature Geosciences) who calculated atmospheric pO_2 based on the $\delta^{13}\text{C}$. We must keep in mind that the main producers of O_2 in the oceans are cyanobacteria and that Earth's atmosphere was most likely O_2 -rich at least since the Proterozoic (see the recent works published by Steadman for the Precambrian and by Brand for the Paleozoic). So, I am not convinced that the uranium isotope record of these carbonates reflects a global redox state of the oceans but rather a specific local environment that was submitted to suboxic or anoxic conditions.

Moreover, I also notice that post-depositional processes such as diagenesis has been eluded with the mention that "despite acquiring a known offset during carbonate diagenesis, it represents a measure of the global oxygenation state of the oceans" on line 102. How you can write that you measure confidently something with an offset? How do you it is a "constant" offset when you are studying Paleozoic sediments and referring to papers that were devoted to modern or very recent carbonate deposits? (Tissot et al. 2018; Chen et al. 2018).

Some sentences are also senseless, see for example line 24: you cannot write that "animal life requires oxygen to SURVIVE".

For all the reasons mentioned above, I cannot recommend publication of this manuscript in its present state, it should be at least deeply modified and revised before publication. You should more put emphasis on the need of cooler marine temperatures allowing more O₂ to be dissolved in water and the availibility of nutrients that is connected to the carbon isotope cycle.

**Stable ocean redox during the main phase of the Great**

**Ordovician Biodiversification Event**

Álvaro del Rey^{1*}, Christian Mac Ørum Rasmussen^{1,2}, Mikael Calner³, Rongchang Wu⁴,

Dan Asael⁵ and Tais W. Dahl¹

¹GLOBE Institute, University of Copenhagen, Øster Voldgade 5-7, DK-1350 Copenhagen

8 K, Denmark

²Natural History Museum of Denmark, Øster Voldgade 5-7, DK-1350 Copenhagen K,

Denmark

³Department of Geology, Lund University, Sölvegatan 12, SE-223 62 Lund, Sweden

⁴Nanjing Institute of Geology and Paleontology, Chinese Academy of Sciences, 39 East

Beijing Road, Nanjing, 210008 China

⁵Department of Earth and Planetary Sciences, Yale University, New Haven, Connecticut

06511, USA

*e-mail: alvarodelrey@science.ku.dk

**ABSTRACT**

The Great Ordovician Biodiversification Event (GOBE) represents the greatest increase in

marine animal biodiversity of the Paleozoic. What caused this transformative evolution of

marine animal life is heavily debated. One hypothesis states that rising atmospheric

oxygen levels drove the biodiversification based on the premise that animal life requires

oxygen to survive. Here, we present uranium isotope data from a marine carbonate

succession spanning the Middle Ordovician that shows the steepest rise in generic richness

and peak of the main phase of the GOBE occurred with global marine redox stability. An
increase in oxygenation ensued later; thus it could not have driven the biodiversification.
Stable marine anoxic zones were the environmental background in the interval leading up
to the maximum increase in biodiversity of the GOBE during which the life expectancy of
evolving genera greatly increased (Dapingian–early Darriwilian). Unstable ocean redox
conditions occurred together with a marine carbon cycle disturbance and a decrease in
relative diversification rates (mid–late Darriwilian). Thus, marine animal life expanded
and lived increasingly longer when marine anoxia was stable over time. Therefore, we
propose that oceanic redox stability was a factor in facilitating the establishment of more
resilient ecosystems allowing marine animal life to radiate.

Keywords: U isotopes, Marine Carbonates, Middle Ordovician, GOBE, Ocean Redox
Stability

**INTRODUCTION**

The Great Ordovician Biodiversification Event (GOBE) was the greatest accumulation of
metazoan richness in Earth’s history (ref.¹ and references therein). Generic richness
increased four-fold during the **Middle–early Late Ordovician** and there was a complete
restructuring of trophic relationships^{2,3}. The first complex marine ecosystems with deeper
tiering, intricate trophic cascades and ecospace partitioning were established⁴. Whereas the
Cambrian–early Ordovician faunas had been dominated by detritus-feeding benthos, from
the mid-Ordovician onwards suspension-feeding organisms vastly radiated not just on the
sea floor but also throughout the water column, as did predation pressure⁵. Reef systems
started resembling modern analogues, transitioning from algal and bacterial mats to ones

with complex frameworks constructed by metazoans⁶. What instigated this transformative
ecological expansion in the marine environment is therefore a fundamental question to
answer. A number of different causal mechanisms have been proposed¹, including two
abiotic factors that have gained momentum. One, climatic cooling inducing a decrease in
tropical sea surface temperatures (SSTs) from Cambrian and Early Ordovician warm
conditions (~42°C) to temperatures similar to present-day (27–32°C) during the Middle
Ordovician Darriwilian Age, is thought to have provided more favorable conditions for
marine animal life thus promoting the radiation^{7–11} (Fig. 1). And two, an increase in
atmospheric oxygenation during the Middle Ordovician, in contrast to concurrent anoxic
events and elevated extinction rates from the late Cambrian to early Ordovician¹², has
been proposed as a driver of the GOBE¹³ (Fig. 1). This is justified either by higher
molecular oxygen (O₂) levels in surface waters fuelling a more energetic lifestyle (e.g.
carnivory) and allowing higher trophic levels, and/or by oxygenating new seafloor areas,
creating new ecospace for marine animals^{14–16}. Because animals depend on O₂ to
survive^{17,18}, a more oxygenated world during the Middle Ordovician might have supported
more animal life. Nevertheless, a recent paleontological study has suggested that there is a
time difference between the onset of the GOBE and the rise in atmospheric O₂¹⁹ and thus,
the role of O₂ as a direct causal mechanism in the expansion of biodiversity remains
unclear.

Here we assess the relationship between oceanic oxygenation and animal radiation during
the GOBE by analyzing uranium (U) isotopes in marine carbonates from a sedimentary
succession spanning the Middle Ordovician (Dapingian–Darriwilian). The U isotope
signature of marine carbonates ($\delta^{238}\text{U}_{\text{CARB}}$) can inform us about the global oxygenation
state of the oceans because the marine U removal in anoxic settings induces a large

positive U isotope offset in sediments deposited under reducing conditions²⁰. In general
terms, fluctuations towards lower $\delta^{238}\text{U}_{\text{CARB}}$ values relative to today ($\delta^{238}\text{U}_{\text{CARB}} = -0.14 \pm$
0.11% , 1sd)²¹ indicates deoxygenation or an expansion of anoxic water masses, whereas
swings towards higher $\delta^{238}\text{U}_{\text{CARB}}$ values similar to modern-day carbonates implies a lesser
extent of anoxia or greater proportion of sediment burial in the oxygenated parts of the
oceans²². Thus, by comparing the U isotope composition of marine carbonates from the
Middle Ordovician with high-resolution global generic richness data, we examine whether
or not changes in the global oxygenation state of the oceans were related to the expansion
of marine animal life of the GOBE.

**BACKGROUND**

**U isotopes in marine carbonates as a proxy for ocean oxygenation**

Microbial U reduction involves large isotope fractionation^{23,24}. Because U(VI) is
predominantly soluble and U(IV) insoluble in the oceans, the reduction process takes U
out of solution and simultaneously fractionates its isotopes with the heavier ²³⁸U isotope
enriched in the product^{23,25}. Therefore, ²³⁸U is preferentially removed in anoxic settings
where U accumulates into the sediments, leaving behind ²³⁸U-depleted (²³⁵U-enriched)
waters. Hence, sediment burial under reducing conditions becomes the main sink of U
associated with isotope fractionation^{20,26}. The incorporation of U into biotic or abiotic
carbonates, although also a large U sink²⁷, only involves limited isotope
fractionation^{21,28,29}. Thus, the extent of anoxic water masses becomes the main control of
the U isotope composition of the oceans²⁵. Lastly, due to its much longer residence time
(~400 kyr) compared to ocean mixing time (~1.5 kyr), U behaves conservatively and has a
globally uniform isotopic distribution in the ocean²⁵. Therefore, marine carbonates formed

under oxic conditions record the isotope composition of seawater at the time of deposition,
which despite acquiring a **known offset** during carbonate diagenesis^{21,30}, it represents a
measure of the global oxygenation state of the oceans²².

**Geological setting**

The analyzed carbonate rocks from the Middle Ordovician consist of a continuous
sedimentary succession from ~~Kinneulle in~~ southern Sweden (Kinnekulle-1 drillcore, Fig.
2A) that represent **cool-water** carbonate facies deposited with limited terrigenous input
within the Baltoscandian Paleobasin³¹ (Fig. 2B). The carbonate succession comprises
mudstones and wackestones with lesser marly limestones and calcareous shales. The basal
part of the studied section (Fig. 3) consists of the Lanna Limestone (Dapingian–lower
Darriwilian (Dw1 stage slice sensu ref.³²)) and is followed by the Holen Limestone (Dw1–
upper Dw2). The Lanna–Holen interval is characterized by a reddish-brown color and a
rather homogeneous macroscopic appearance. Within the lower Holen Limestone, a
conspicuous and discrete gray limestone interval is referred to as the Täljsten³¹. The gray-
colored Skövde Limestone (middle Dw3) disconformably overlies the Holen Limestone,
evidencing a hiatus at Kinnekulle encompassing the lower Dw3³¹. The top of the studied
section comprises the **gray basal beds** of the Gullhögen Formation (upper Dw3) deposited
conformably over the Skövde Limestone.

**RESULTS**

~~A $\delta^{238}\text{U}$ record through the Middle Ordovician~~

The $\delta^{238}\text{U}$ curve from the carbonate succession of Kinnekulle (Fig. 3) exhibits secular
trends that can be subdivided into two distinct intervals depending on the amplitude or
stability of their $\delta^{238}\text{U}$ variations (Fig. 3). In the lower part of the section up until the
Täljsten (Dp1–Dw1), $\delta^{238}\text{U}$ trends are stable with a lower average value and similar
variance compared to modern carbonates at $-0.36 \pm 0.09\text{‰}$ (1sd, $n = 26$). The most stable
or invariant $\delta^{238}\text{U}$ trend occurs from middle Dp3 to upper Dw1 (from -0.53 to -0.33‰). A
second interval then follows (Dw2–Dw3) with large amplitude fluctuations or unstable
$\delta^{238}\text{U}$ (from -0.78 to 0.16‰), beginning in the Täljsten with an abrupt swing in $\delta^{238}\text{U}$
values (-0.78 to -0.25‰). From this point on, a positive $\delta^{238}\text{U}$ excursion is observed
during the Dw2, followed by a negative one with a large point-by-point variability near
the unconformity at the Dw2–Dw3 boundary. A large positive, yet highly fluctuating
$\delta^{238}\text{U}$ trend indicative of a change towards more oxygenated ocean conditions occurs from
the Dw3.

Elemental data shows elevated U content in the gray facies of the Täljsten and Skövde
Limestone–Gullhögen Formation associated with higher $\delta^{238}\text{U}$ variability. The secular
$\delta^{238}\text{U}$ changes are not coupled to variations in elemental ratios (Mg/Ca, Sr/Ca, Mn/Sr,
Al/Ca, Mo/Ca) (Supplementary Fig. 1), indicating that the $\delta^{238}\text{U}$ trends are not
systematically related to changes in primary carbonate mineralogy, chemical alteration by
diagenetic fluids, terrigenous input, or local redox conditions (see Supplementary
Information for all isotopic and geochemical data and further discussion about diagenesis).

**DISCUSSIONS**

~~The global oxygenation state of the oceans and changes in marine biodiversity~~

The $\delta^{238}\text{U}$ record across the Dapingian and Darriwilian shows a change from an interval
with a relatively stable global ocean oxygenation state (particularly from middle Dp3–
upper Dw1) to unstable redox conditions characterized by large fluctuations in $\delta^{238}\text{U}$
values (Dw2–Dw3) (Fig. 3). A continuous shift towards more oxygenated oceans is first
seen in the late Darriwilian (Dw3). Thus, an increase in oxygenation could not have
triggered the earliest Darriwilian inception of the main phase of the GOBE as previously
suggested¹³. Both global and regional richness data shows that the peak of this main phase
occurred during the late Dw1^{1,19,33}. Therefore, we find that a relatively stable global
oxygenation state of the oceans prevailed during the time period leading up to the
maximum increase of biodiversity of the GOBE (Interval I, Fig. 3).

After this stable $\delta^{238}\text{U}$ phase, biodiversity and relative diversification and origination rates
decrease when large $\delta^{238}\text{U}$ fluctuations are observed (Interval II, Fig. 3). This change
coincides with the onset of a perturbation of the global marine carbon cycle known as the
“Middle Darriwilian Isotope Carbon Excursion” or MDICE (from ca. early Dw2)^{34–36}.
Thus, relative diversification slows down when the redox landscape becomes unstable and
the carbon cycle is globally disturbed. Further, extinction rates increase and brachiopod
richness reaches a low when the $\delta^{238}\text{U}$ curve suggests a shift towards more anoxic water
masses in the oceans, around the unconformity of the section (Dw2–Dw3 boundary) (Fig.
3).

The transition from a relatively stable to unstable redox landscape and accompanying
changes in marine animal biodiversity occurs in the conspicuous Täljsten interval within
the lower Hølen Limestone near the Dw1–Dw2 boundary (Fig. 3). The Täljsten possess a
series of atypical facies and paleontological features that evidence a period of stressed

environments, including opportunistic disaster taxa suggesting the collapse of marine
ecosystems^{31,37}. We see this expressed in a rapid and large $\delta^{238}\text{U}$ fluctuation within these
beds and elevated abundances of redox sensitive elements (U/Ca, Mo/Ca, P/Ca,
Supplementary Fig. 2), which suggest reducing conditions in the local depositional
environment³⁷. In addition, a similar lithological shift is also observed in coeval
limestones in South China^{38,39}. Therefore, the abrupt lithological shift in the rather uniform
carbonate succession in two distinct paleocontinents suggests the transition was initiated
by a global event that disrupted sedimentary cycles and the marine redox landscape.

**Animal life thrived during the Dapingian–early Darriwilian**

An important factor that has been causally related with the onset of the GOBE is cooling
oceans reaching tropical SSTs similar to present day in the Middle Ordovician, providing
more favorable conditions for marine animal life^{7,11}. Equatorial modern-like SSTs were
attained from around the late Dapingian⁷ and a significant 4–5°C cooling affecting the
Baltoscandian Paleobasin was identified in strata just above the Dapingian–Darriwilian
boundary¹¹. Thus, ocean cooling occurred in the same time interval with the comparatively
most stable ocean redox conditions and the maximum increase in marine biodiversity
(middle Dp3–upper Dw1).

During the Dapingian–early Darriwilian, the life expectancies of genera reached an early
Paleozoic maximum just prior to the onset of the GOBE³³. Thus, generic lineages survived
longer at the time with the most stable $\delta^{238}\text{U}$ trends (Fig. 3). Long-life expectancies
indicate a long persistence of the ecologic relationships among genera or ecosystem
resilience, and processes that increased levels of ecosystem resilience were major factors

of marine biodiversity accumulation³³. Therefore, we suggest ecosystems evolved and
became more resilient, persisting for longer time spans, simply because animal life thrived
when the marine redox landscape was stable and tropical ocean temperatures were more
favorable (Fig. 3).

CONCLUSIONS

The comparison between the $\delta^{238}\text{U}$ of marine carbonates and marine animal biodiversity
through the Middle Ordovician shows that the inception and main phase of the largest
animal radiation of the Paleozoic—the GOBE—was not linked to an increase in ocean
oxygenation, but to a rather stable marine redox landscape. An oxygenation event in the
oceans (this study) and atmosphere¹³ is recorded after and therefore, it could not have
driven the biodiversification event. The GOBE is also marked by a notable peak in the life
expectancies of genera just before the onset of the radiation and in the time interval with
the most stable $\delta^{238}\text{U}$ trends (Dapingian–early Darriwilian), prior to a global disturbance
of the marine carbon cycle (pre MDICE) and when cooling oceans reached modern-like
equatorial SSTs. Therefore, we suggests that the relative stability of the marine redox
landscape over geological timescales (>1 Myr) was one of the fundamental factors
allowing animal life to simply establish itself in its environment for longer times by
preventing the expansion of anoxia to disturb marine ecosystems on shorter timescales like
it occurred during the late Cambrian–early Ordovician. This created favorable conditions
for marine animal life, which together with cooler oceans were likely key in promoting the
greatest animal radiation in Earth’s history.

METHODS

Analyzed samples for U isotopes correspond to ca. 1 g powdered carbonate rock each.
Powders were extracted from the rocks' matrices wherever possible, avoiding large grains
and areas that macroscopically evidence secondary alteration (e.g., weathering, veins,
etc.). Samples were dissolved using a soft leaching procedure using ca. 45 mL 0.5 M HCl
to avoid the dissolution of any non-carbonate fraction, but achieving complete dissolution
of the carbonates²². A small aliquot (0.15%) of the leachate was taken for major and trace
element concentrations analyses and the rest (99.85%) was used for uranium isotope
analyses.

Major and trace element concentrations were analyzed using high-resolution sector field
inductively coupled plasma mass spectrometry (ICP-MS), ThermoFisher Scientific
Element XR, at the Yale Metal Geochemistry Center, Yale University. Measurement
precision was generally better than 5% (2σ). BVO-2 and NOD-A-1 USGS geostandards
were processed along with samples in each run using a standard sample introduction
system⁴⁰ and were within $\pm 5\%$ of reported values.

For U isotope analyses, U was extracted and purified from the carbonate leachates using
the UTEVA ion exchange resin procedure and measured using the ^{233}U - ^{236}U double-spike
method⁴¹⁻⁴³ on a ThermoFisher Scientific Neptune Plus Multicollector ICP-MS at the Yale
Metal Geochemistry Center, Yale University. Up to 150 ng U per sample were purified
passing through UTEVA ion exchange resin twice after adding 240 μL double-spike
solution per 150 ng U to have the same sample/spike ratio in all samples and standards.
The IRMM-3636 ^{233}U - ^{236}U double-spike method is used to correct for instrumental mass
bias and potential isotopic fractionation during column chemistry⁴¹. In summary, after

adding the spike, samples were dissolved in 7 M HNO₃ and then loaded into 1 mL
UTEVA-resin-loaded columns pre-cleaned with 8 mL 0.05 M HCl and pre-conditioned
with 7 M HNO₃. Matrix elements (except thorium) were eluted using 15 mL 3 M HNO₃
and Th using 6 mL 5 M HCl. U was then released from the resin using 13 mL 0.05 HCl.
The final solutions with collected U were then dried down and treated with small amounts
of H₂O₂ and concentrated HNO₃ to eliminate potential residual organic compounds
leached from the resin. This procedure was applied twice and after the second time,
samples were dried down and brought up in 3 mL 5% HNO₃ (50 ppb U) for isotope
composition analyses. Samples were introduced to the multicollector ICP-MS using an
ApexIR sample introduction system and measured at low resolution. Isotopes ²³²Th, ²³³U,
²³⁵U, ²³⁶U and ²³⁸U were measured simultaneously on Faraday collectors connected to 10¹¹
Ω amplifiers achieving a signal of ~ 25–45 volts on ²³⁸U, depending on concentration.
Procedural blanks were 10-40 pg U. The data are reported as parts per thousand (‰)
deviation of sample's ²³⁸U/²³⁵U ratio relative to the CRM-145 reference standard (δ²³⁸U).
Accuracy and external reproducibility of the data were evaluated by analyzing Ricca and
CRM-129a reference materials processed through the same chemical purification
procedure with the same sample/spike ratio as the studied samples. External precision
(1STD) was better than 0.07‰.

**ACKNOWLEDGMENTS**

We thank Xuefang Wu for laboratory assistance at Yale University during the COVID-19
pandemic. We also thank Anders Lindskog for his work and discussions on the conodont
biostratigraphy. AdR and TWD acknowledge financial support from the Carlsberg
Foundation through its Distinguished Associate Professor program to TWD (CF16–0876).

TWD thanks the Danish Council for Independent Research (7014-00295B and 8102-
00005B). CMØR is grateful for funding from GeoCenter Denmark (2015-5 and 3-2017).
This is a contribution to the IGCP Project 653 ‘The Onset of the Great Ordovician
Biodiversification Event’. MC and RW acknowledge the Crafoord foundation (20140806)
and the Royal Physiographic Society of Lund, respectively, for funding the recovery of the
Kinnekulle-1 drillcore.

**AUTHOR CONTRIBUTIONS**

This study was designed by TWD with input from MC and CMØR. Sampling and isotope
analyses preparation done by AdR. Isotope measurements made by DA. Conodont
biostratigraphy made by RW. AdR and TWD wrote the paper with input from all co-
authors.

**COMPETING INTERESTS**

The authors declare no competing financial interests.

**FIGURES**

[revised manuscript text omitted]

Middle Ordovician ca. 465 Ma

Review Report of del Rey et al. (COMMSENV-21-0778-T)

Summary

This study presents a new set of U isotope data across the Great Ordovician Biodiversification Event (GOBE) from the carbonate succession of Kinnekulle. In contrast to previous studies, which attributed the GOBE to an increase in atmospheric oxygenation during Middle Ordovician, this study concludes that the inception and main phase of the GOBE were linked to a stable marine redox landscape. Thus, the author suggests the relative stability of the marine redox landscape over geological timescales over 1 Myr should be considered as one of the fundamental factors for animal radiation events.

Overall Comments

Strengths: The paper is straightforward with a nice and simple narrative. The major novelty is that this work evaluated the relationship between marine oxygenation and animal radiation by examining the fluctuation of oceanic anoxia throughout geological time rather than by determining the absolute extent of oceanic oxygenation at the time of biodiversity expansion.

Weaknesses:

1. Although the authors acknowledged the documented impact of carbonate diagenesis in the supplementary, the variable nature of this impact ($0.27 \pm 0.28 \text{ ‰}$) is seldom considered in the main text. This is problematic because diagenesis might not always systematically shift the $\delta^{238}\text{U}$ values during the middle Ordovician, and changes in the depositional environment can change the magnitude of diagenetic offset (Zhang et al., 2020). This might substantially influence the apparent presence or lack of $\delta^{238}\text{U}$ fluctuation by either amplifying or reducing the noise variability of the data set. Combined with the lower data density in the GOBE (relative to the period just before it), it is unclear whether the apparent stability of the signal is real or an artefact of lower number of data points and changes in deposition environment.
2. While the paper provides a detailed description of the sample lithology, there is not enough information on the depositional environment. The assumption of a constant offset between $\delta^{238}\text{U}_{\text{CARB}}$ and $\delta^{238}\text{U}_{\text{SW}}$ would be even less valid if the carbonates were not deposited in an open ocean environment (Clarkson et al., 2018; Tissot et al., 2018).

Although the influence of factors mentioned above is still an ongoing topic for U isotope studies, it would serve the main conclusion of the paper if a more thorough evaluation of these effects was performed. It is also, surprising that the authors do not attempt to provide a quantitative estimate of seafloor anoxia during the GOBE, as numerous papers have reviewed typical models that can be used (Lau et al., 2016, 2017; Zhang et al., 2020) and more recent studies have even provided interesting and quantitative approaches to properly consider the impact of these offsets. For example, Monte Carlo simulation can quantify uncertainties introduced by early diagenetic alterations (Pimentel-Galvan et al., 2022; Kipp and Tissot, 2022).

Provided the above main comments are properly addressed, I think the manuscript can would be acceptable for publication in Communications Earth & Environment.

Minor Problems

Main text

L126. “Secular trends ... $\delta^{238}\text{U}$ variations”

Because the division of different intervals based on amplitude or stability of $\delta^{238}\text{U}$ is a key concept in this paper, it would be useful if a more detailed/quantitative description on the criteria used to distinguishing these intervals were provided.

L238. There is a typo on the name of geostandard (BHVO-2 rather than BVO-2)

L265. For checking external reproducibility with Ricca and CRM-129a, please provide $\delta^{238}\text{U}$ of these standards.

Figure 3.

- For the U/Ca plot, the difference of the points near 0 can be better resolved with log scale.
- Make the labelling of interval I and II clearer. It is difficult to find them in the figure now.

Supplementary information

Table 1. Please provide the unit for concentration ratios.

Figure 1. It would be more informative and diagnostic to plot $\delta^{238}\text{U}$ vs Al/U when trying to tease out detrital contributions to the data, as in such a space, detrital contamination with a single end member would be a straight line. I couldn't test this hypothesis, as there are not enough significant digits on the Al/C ratios and no Al/U ratios are reported.

Reference:

- Clarkson M. O., Stirling C. H., Jenkyns H. C., Dickson A. J., Porcelli D., Moy C. M., Von Strandmann P. P. A. E., Cooke I. R. and Lenton T. M. (2018) Uranium isotope evidence for two episodes of deoxygenation during Oceanic Anoxic Event 2. *Proc. Natl. Acad. Sci.* 115, 2918–2923.
- Kipp M. A. and Tissot F. L. H. (2022) Inverse methods for consistent quantification of seafloor anoxia using uranium isotope data from marine sediments. *Earth Planet. Sci. Lett.* 577.
- Lau K. V., Maher K., Altiner D., Kelley B. M., Kump L. R., Lehrmann D. J., Silva-Tamayo J. C., Weaver K. L., Yu M. and Payne J. L. (2016) Marine anoxia and delayed Earth system recovery after the end-Permian extinction. *Proc. Natl. Acad. Sci. U. S. A.* 113, 2360–2365.
- Lau K. V., Macdonald F. A., Maher K. and Payne J. L. (2017) Uranium isotope evidence for temporary ocean oxygenation in the aftermath of the Sturtian Snowball Earth. *Earth Planet. Sci. Lett.* 458, 282–292. Available at: <http://dx.doi.org/10.1016/j.epsl.2016.10.043>.
- Pimentel-Galvan M., Lau K. V., Maher K., Mukerji T., Lehrmann D. J., Altiner D. and Payne J. L. (2022) Duration and Intensity of End-Permian Marine Anoxia. *Geochemistry, Geophys. Geosystems* 23, 1–19.
- Tissot F. L. H., Chen C., Go B. M., Naziemiec M., Healy G., Bekker A., Swart P. K. and Dauphas N. (2018) Controls of eustasy and diagenesis on the $^{238}\text{U}/^{235}\text{U}$ of carbonates and evolution of the seawater ($^{234}\text{U}/^{238}\text{U}$) during the last 1.4 Myr. *Geochim. Cosmochim. Acta* 242, 233–265.
- Zhang F., Lenton T. M., del Rey Á., Romaniello S. J., Chen X., Planavsky N. J., Clarkson M. O., Dahl T. W., Lau K. V., Wang W., Li Z., Zhao M., Isson T., Algeo T. J. and Anbar A. D. (2020) Uranium isotopes in marine carbonates as a global ocean paleoredox proxy: A critical review. *Geochim. Cosmochim. Acta*.

Reviewers' comments and authors' responses Response to Reviewer 1 (Maya Elrick)

Reviewer 1 Comment 1

The del Rey et al manuscript is a straightforward, well organized study testing the hypothesis that a global seawater oxygenation increase triggered the GOBE. The authors use $\delta^{238}\text{U}$ trends from Swedish marine carbonates and comparisons to previously reported biodiversity, origination/extinction data sets to address the question. The new $\delta^{238}\text{U}$ data spans a relatively short time interval in the Middle Ordovician (~12 My) and assumes that the main onset of GOBE biodiversity increase occurs near the Dapingian-Darriwilian boundary. The entire hypothesis of the study is premised on assuming the GOBE onset occurred during this short time interval. However, the results of a recent study by Servais et al (2021) suggest that this assumption requires re-evaluation because the various, widely used, biodiversity data sets are biased and/or incomplete. Instead, Servais et al argue that the Ordovician radiation was a complex, long-term process of multiple marine diversifications and that the idea of a relatively short, dramatic global radiation at a particular time interval is an oversimplification--neither was it short or coevally global, and the biodiversity data sets used in the past are biased and/or incomplete. Servais et al argue that depending on which biodiversity data set is used, the GOBE occurred anytime throughout the Early to Late Ordovician (Servais' et al figure 1). Given this, the key interpretation of this study of the timing of the GOBE onset is associated with stable seawater redox conditions rather than increasing $\delta^{238}\text{U}$ trends (indicating increasing oxygenation) is problematic. The authors need to clearly address this issue, provide more explanation on the GOBE timing (and why they choose to use particular timing interpretations) and provide solid evidence to refute the Servais et al findings if they disagree.

Response: We thank the reviewer for clear summary of our work, and also for the constructive comment regarding hypothesis testing for animal radiation during ~12 Myrs of the Mid-Ordovician.

Reviewer 1 is correct that Servais et al. (2016, 2021), Servais & Harper (2018) and Harper et al. (2020) argue that the GOBE should be viewed as a prolonged phase spanning the entire Ordovician and with its roots in the late Cambrian. However, their claim is completely unsupported by any evidence. This we have tried to outline below.

Opposed to the view advanced by the Servais–Harper papers stands the bulk of new and older research on the timing of the GOBE across geographical regions and across clades to come out since Webby et al. 2004 first coined the term GOBE (see, for instance, Miller & Foote 1996; Rasmussen et al., 2007; Trotter et al 2008; Harper et al., 2013; Adrain et al 2013; Trubovitz & Stigall 2016; Hints et al., 2018; Cole 2018; Colmenar & Rasmussen 2018; Ernst 2018; Rasmussen et al., 2019; Kröger et al., 2019 and many more). Add to this, that the original Sepkoski dataset show the same trend of a Mid Ordovician diversity increase (Sepkoski 1979; 1981; 1995; Sepkoski & Sheehan 1983). We particularly base our conclusions on the most recent global estimates as they are highly resolved datasets backed up by rigorous statistical evidence (Kröger et al. 2019; Rasmussen et al. 2019). They show a clear rise in biodiversity accumulation during the earliest Middle Ordovician (Dp3–Dw2 timeslices).

In the new study by Servais et al. 2021 they again argue for a prolonged rise in richness by comparing already published studies (see their figure 1). However, they show no new data at all to back up their claims. Only comparisons of other studies which supposedly should show different richness trends (Servais et al., 2021, fig. 1). This comparison completely ignores that recent studies are of much higher temporal resolution than the original Sepkoski dataset. Despite of this, Servais et al. 2021 figs 1a–c still clearly shows the same middle Ordovician event (that they try to argue against...). Only 1d, which is data from peri-Gondwana, show an earlier rise in biodiversity, which is already well known and described by several studies (see, for instance, Stigall et al 2019.). This apparent diachroneity between South China and the rest of the world was most recently shown to be much

narrower than previously thought with the Chinese species richness data now also peaking during the earliest Darriwilian (Deng et al., 2021).

So, it is indeed the claims of Servais et al. 2021 that represents a provocative oversimplification of the biodiversity data that already exists in the literature. Their argumentation of a prolonged rise in richness can only be supported by broader binned datasets that are not only misleading, but also blurs the richness signal (see, for instance, Harper et al., 2020). An example is that their argumentation for a prolonged rise in richness from the Cambrian through to the Late Ordovician completely ignores that when the data is binned at high temporal resolution, it is only the CR-method used by the Kröger and Rasmussen studies that predicts even just moderate richness levels in the late Cambrian – all other methods show a considerable drop in diversity during the last 10 myr of the Cambrian period (see Rasmussen et al., 2019, Supp. File). The CR-method clearly shows the dramatic Mid Ordovician rise that follows after a nearly 50 myr long richness plateau (where Servais-Harper argue for continued diversity rise). So, even though they try to argue against the result of the CR-method, they are actually heavily dependent on it if they are to back up their claims of a consistent rise in diversity during the late Cambrian. In any case, a broader temporal binning is not useful when species originate or go extinct within the scales of millennia – one could also make a study with just two bins: a Precambrian and a Phanerozoic bin. In such a case one would see a dramatic increase in richness at the Precambrian–Cambrian boundary, but that would obviously not inform us about the detailed timing of animal radiation.

In addition to the above, the timing of the GOBE has been exhaustively reviewed by Stigall et al. (2019) which show the GOBE should be viewed as a Middle Ordovician ‘event’, not a protracted phase spanning 40 million years. That said, some clades do show a gradual rise earlier in the Ordovician and some originate during the late Cambrian, but neither of these can be characterized as other than background speciation, something that occurs basically throughout the Phanerozoic. The only exception globally from a Middle Ordovician main phase of the GOBE is in South China where a pre-GOBE radiation commenced during the Early Ordovician (Floian) and this may well have sparked the GOBE itself (see Zhan & Harper, 2006; Rong et al., 2007; Fan et al., 2020). But, this regional radiation cannot be deemed a global event.

As all this knowledge has been completely ignored by Servais et al. 2021 and as that study does not bring forward any new data to the discussion we see no point in mentioning it, nor deviate from the common view of the bulk of the literature.

Reviewer 1 Comment 2

Below are specific line-by-line comments.

Line 44—This is the first time the (assumed) GOBE timing is mentioned. A clearer discussion of the radiation and its timing is necessary as the entire study hinges on this.

Response: Yes. We specified that the study concerns the main phase of the GOBE. L71ff now reads:

Here we assess the relationship between oceanic oxygenation and animal radiation during a ~12 million year-long time interval during the main phase of the GOBE by analyzing uranium (U) isotopes in marine carbonates from a sedimentary succession deposited in the Middle Ordovician (Dapingian–Darriwilian).

Reviewer 1 Comment 3

Lines 66-69- The second potential driver (Middle Ordovician oxygenation) is provided then an argument on why it is not supported is given. This same pattern should be provided for the first hypothesis (SST cooling) and why is or is not supported.

Response: Corrected. We added a supporting clause:

L54ff now reads:

One, climatic cooling inducing a decrease in tropical sea surface temperatures (SSTs) from Cambrian and Early Ordovician warm conditions (~42°C) to temperatures similar to present-day (27–32°C) during the Middle Ordovician Darriwilian Age, is thought to have provided more favorable conditions for marine animal life thus promoting the radiation⁷⁻¹¹(Fig. 1).

Reviewer 1 Comment 4

Lines 73-81—this brief discussion of U isotopes as redox proxy is more appropriate in Background rather than in Introduction.

Response: Yes. We moved this paragraph down to Background.

Reviewer 1 Comment 5

Line 76- be specific in stating a large fractionation occurs in euxinic sediments specifically, not general anoxic sediments.

Response: Yes. We corrected this in line 83:

The U isotope signature of marine carbonates ($\delta^{238}U_{CARB}$) can inform us about the global oxygenation state of the oceans because the marine U removal in anoxic and sulfidic (euxinic) settings induces a large positive U isotope offset in sediments²⁰,

Reviewer 1 Comment 6

Line 77-78—not clear what modern values of -0.14 permil is referring to

Response: Corrected. Line 86 now reads:

“lower $\delta^{238}U_{CARB}$ values than modern carbonate sediments ($\delta^{238}U_{CARB} = -0.14 \pm 0.11\text{‰}$, 1sd)²¹”

Reviewer 1 Comment 7

Line 93- euxinic specifically

Response: Corrected.

Reviewer 1 Comment 8

Line 102- 'known' is too optimistic, the 'most recently documented range of diagenetic offset' is better language

Response: Corrected.

Reviewer 1 Comment 9

Line 109- readers outside the carbonate community do not know what cool water carbonates means

Response: We provide reference to Lindskog & Eriksson 2017 and Jaanusson 1973, who gives a careful description of cool water carbonate facies in Baltica at this time. We keep the description "cool-water carbonate facies" believe the sentence gives an accurate picture of the type of deposition environment also to the non-experts.

Reviewer 1 Comment 10

Line 119- need rock type information rather than just 'beds'

Response: Corrected. Line 120ff now reads:

The top of the studied section comprises the gray basal marly/nodular limestone of the Gullhögen Formation (upper Dw3) deposited conformably over the Skövde Limestone.

Reviewer 1 Comment 11

Line 127- 'variations' cannot be stable, reword

Response: Corrected. Line 130 now reads:

$\delta^{238}\text{U}$ trends are steady

and Line 131 now reads:

The smallest $\delta^{238}\text{U}$ variance occurs from the middle Dp3 to the upper Dw1

Reviewer 1 Comment 12

Line 128-129 - Confusing sentence, average measured values may be similar to modern carbonates, but the variance is not similar. Needs clarification.

Response: Corrected. Line 129 now reads:

In the lower part of the section up until the Täljsten (Dp1–Dw1), $\delta^{238}\text{U}$ trends are steady with a lower average value and similar or even smaller variance than observed in modern carbonates at $-0.36 \pm 0.09\text{‰}$ (1sd, n = 26).

Reviewer 1 Comment 13

Line 132-133- Best not to focus on the likely outliers within the Täljsten (for example similar point-by-point variability at ~3 m in Dp2). These large point-by-point variabilities at the sub-meter scale of $>\sim 0.2\text{--}0.3$ permil are not realistic for the global oceans. Focus instead on the major and longer term excursions starting above Täljsten.

Response: Corrected. We focus on the main trends and added the fluctuations second. Line 132ff now reads:

The following interval (Dw2–Dw3) shows unstable $\delta^{238}\text{U}$ values with large amplitude fluctuations (from -0.78 to 0.16‰). A positive $\delta^{238}\text{U}$ trend is observed during the Dw2, followed by a negative one with a large point-by-point variability in the Täljsten and near the unconformity at the Dw2–Dw3 boundary.

Reviewer 1 Comment 14

Line 139- define gray facies

Response: Corrected. We added “gray limestone facies”. Ref 31 gives further details.

Reviewer 1 Comment 15

Line 142—statement is too optimistic, best to say they are not significantly altered by....

Response: Corrected. We added “gray limestone facies”. Ref 31 gives further details.

Reviewer 1 Comment 16

Line 154-157—see comments above about GOBE timing, this statement is based on the assumption that the main GOBE increase occurred in this relatively short time interval

Response: Corrected. We changed the wording to “Thus, an increase in oxygenation could not have triggered the earliest Darriwilian inception during this phase of the GOBE, as previously suggested¹³.”

Reviewer 1 Comment 17

Line 159- Interval I is not labelled on Figure 3 or mentioned in figure caption.

Response: Corrected. We added the following text to the caption of figure 3:

Intervals I and II are shown next to the $\delta^{38}U$ data.

Reviewer 1 Comment 18

Line 161-169—entire discussion is based on assumption that diversity and extinction/origination rate trends of previous studies are correct and does not include Servais et al 2021 results.

Response: See our exhaustive reply to Reviewer 1’s comment 1. We may add that particularly when it comes to speciation rates this has been well covered both by C.M.Ø. Rasmussen et al. 2007; J.A. Rasmussen et al 2021; Trubovitz & Stigall 2018; Kröger et al. 2019; Deng et al 2021 but see also older literature, for instance, Miller & Foote 1996 – all studies that clearly show an extraordinarily high accumulation of species across clades during the earliest Darriwilian (see a more exhaustive list of references in Stigall et al., 2019). This massive amount of evidence for a mid-Ordovician event – that is often referred to as the main phase of the GOBE – cannot be questioned by one study (Servais et al 2021) that merely compares published richness curves constructed using different statistical methods and temporal binning but without presenting any counter evidence in the form of new data.

Reviewer 1 Comment 19

Line 175—opportunistic disaster taxa needs explanation, what are they, what do they suggest and why?

Response: In the Täljsten level widespread coquinas of the echinoderm genus *Eosphaeronites* is found (known locally as the Sphaeronites bed). The same is, for instance, seen with orthoceratites that occur abundantly in almost rock-forming quantities. This bears witness of an opportunistic lifestyle terminated by disaster events. We modified line 280ff, so it now reads:

The Täljsten possess a series of atypical facies and paleontological features that evidence a period of stressed environments, including opportunistic disaster taxa., such as mass occurrences of the echinoderm genus Eosphaeronites, suggesting the collapse of marine ecosystems^{37,47}

Reviewer 1 Comment 20 and 21

Line 181—to emphasis this, provide lateral distance between the two continents recording this event

Line 186—when did GOBE really start?

Response: Longitudinal paleo-distances are not well constrained, but paleomap reconstructions suggest China and Baltica were at opposite sides of the planet (Scotese, 2013). To clarify this, we added “far apart” in line 179ff. Also, “transition” does not refer to GOBE, but to the environmental shift from stable to unstable redox landscape. We clarified this, so line 179ff now reads:

Therefore, the abrupt lithological shift in the rather uniform carbonate succession in two distinct paleocontinents far apart suggests the transition from stable to unstable conditions was initiated by a global event that disrupted sedimentary cycles and the marine redox landscape.

Reviewer 1 Comment 22

192—see previous comments on GOBE timing

Response: We clarified where the rapid diversification is recorded. Line 306ff now reads:

Thus, ocean cooling occurred in the same time interval with the comparatively most stable ocean redox conditions and the maximum increase in marine biodiversity accumulation (middle Dp3–upper Dw1).

Reviewer 1 Comment 23

Line 202-203—be specific on what SST define ‘more favorable’. Cooler, warmer, transitioning?

Response: Correct. We give an example of how lower temperatures may have contributed to animal diversification via a greater metabolic index (or “scope”) in larger parts of the oceans.

Therefore, we suggest ecosystems evolved and became more resilient, persisting for longer time spans, simply because animal life thrived when the marine redox landscape was stable and tropical ocean temperatures were lower and more favorable, providing a higher metabolic index for animals in greater parts of the oceans^{L2} (Fig. 3).

Reviewer 1 Comment 24

Line 208-209—see GOBE timing comments

Response: Corrected. We clarified that our study refers to the main phase of the GOBE. Line 211ff now reads:

An oxygenation event in the oceans (this study) and atmosphere¹³ is recorded after and therefore, it could not have driven this phase of the biodiversification event.

Reviewer 1 Comment 25

Line 216-221- run on sentence and confusing concepts, rewrite

Response: Corrected. The sentence now reads:

This phase is also marked by a notable peak in the life expectancies of genera when $\delta^{238}\text{U}$ showed the most stable trend (Dapingian–early Darriwilian), equatorial seawater temperatures had reached modern-like SSTs and the marine carbon cycle was also rather stable (prior to the global disturbance recorded by the MDICE).

Reviewer 1 Comment 26

Figure 1-3. Font size on all figures need enlargement

Response: Corrected. The minimum font size is 8 pt as recommended in the guidelines.

Reviewer 1 Comment 27

Figure 3. Pink band not labelled and Interval I mentioned in text is not shown

Response: The pink band and Interval I and II have now been clarified in the caption of figure 3.

Response to Reviewer 2 (anonymous)

Reviewer 2 Comment 1

This study presents a new set of U isotope data across the Great Ordovician Biodiversification Event (GOBE) from the carbonate succession of Kinnekulle. In contrast to previous studies, which attributed the GOBE to an increase in atmospheric oxygenation during Middle Ordovician, this study concludes that the inception and main phase of the GOBE were linked to a stable marine redox landscape. Thus, the author suggests the relative stability of the marine redox landscape over geological timescales over 1 Myr should be considered as one of the fundamental factors for animal radiation events.

Overall Comments

Strengths: The paper is straightforward with a nice and simple narrative. The major novelty is that this work evaluated the relationship between marine oxygenation and animal radiation by examining the fluctuation of oceanic anoxia throughout geological time rather than by determining the absolute extent of oceanic oxygenation at the time of biodiversity expansion.

Response: Indeed, we thank the reviewer for an exact summary of our manuscript. Our point is rather simple.

Reviewer 2 Comment 2

Weaknesses:

1. Although the authors acknowledged the documented impact of carbonate diagenesis in the supplementary, the variable nature of this impact (0.27 ± 0.28 ‰) is seldom considered in the main text. This is problematic because diagenesis might not always systematically shift the $\delta^{238}\text{U}$ values during the middle Ordovician, and changes in the depositional environment can change the magnitude of diagenetic offset (Zhang et al., 2020). This might substantially influence the apparent presence or lack of $\delta^{238}\text{U}$ fluctuation by either amplifying or reducing the noise variability of the data set. Combined with the lower data density in the GOBE (relative to the period just before it), it is unclear whether the apparent stability of the signal is real or an artefact of lower number of data points and changes in deposition environment.

Response: Yes, this is a valid concern. Interval I and II contains 20 and 35 samples, respectively. There are more than 100 data points from modern carbonates. We have now calculated the statistical significance of the observed $\delta^{238}\text{U}$ variability and compared it to that of modern Bahamian carbonates using a statistical F-test of equal variances. Interval I has significantly lower variance than modern carbonates ($p < 0.05$), whereas Interval II has significantly larger variance. We added a sentence in L. 133-137 to introduce the significance of the observed variability in Interval I and II, and clarified why the larger variability observed in interval II is ascribed to secular changes in seawater composition:

A statistical F-test demonstrates significantly smaller variance in Interval I ($p < 0.05$) and larger variance in interval II ($p < 10^{-4}$) than in modern Bahamas carbonates that have not been altered by meteoric diagenesis (Table S2). Therefore, the systematic variability of the $\delta^{238}\text{U}$ curve in Interval II may reflect secular changes in the composition of global seawater.

Reviewer 2 Comment 3

While the paper provides a detailed description of the sample lithology, there is not enough information on the depositional environment. The assumption of a constant offset between $\delta^{238}\text{UCARB}$ and $\delta^{238}\text{USW}$ would be even less valid if the carbonates were not deposited in an open ocean environment (Clarkson et al., 2018; Tissot et al., 2018).

Response: Corrected. Line 127ff now reads:

The depositional environment for the Lanna and Holen limestones was an open marine, fairly deep setting with low depositional rates (3-5 mm/kyr)^{40,38} and recurrent episodes of non-deposition and formation of corrosional hardgrounds. The erosional surface and associated hiatus at the top of the Holen limestone attests to a longer period of erosion and non-deposition. The alternating red or gray sediment colour may have been caused by sea-level change through changing accumulation rates and early diagenetic redox potential⁴¹.

Reviewer 2 Comment 4

Although the influence of factors mentioned above is still an ongoing topic for U isotope studies, it would serve the main conclusion of the paper if a more thorough evaluation of these effects was performed. It is also, surprising that the authors do not attempt to provide a quantitative estimate of seafloor anoxia during the GOBE, as numerous papers have reviewed typical models that can be used (Lau et al., 2016, 2017; Zhang et al., 2020) and more recent studies have even provided interesting and quantitative approaches to properly consider the impact of these offsets. For example, Monte Carlo simulation can quantify uncertainties introduced by early diagenetic alterations (Pimentel-Galvan et al., 2022; Kipp and Tissot, 2022).

Response: Indeed, we intentionally do not provide estimates for the absolute coverage of seafloor euxinia because it distracts our main point: stability during diversification. The stability is significant during the main phase of the GOBE diversification regardless of the absolute extent of anoxic and sulfidic water masses. That said, other papers have provided estimates that are consistent with our results, so we added a sentence with reference to these works. Line 154ff now reads:

A lower $\delta^{238}\text{U}$ value of Mid-Ordovician carbonates relative to modern carbonates points to a generally more oxygen-depleted ocean with greater proportion of euxinic U burial than today, not too dissimilar from the Late Ordovician and late Silurian ocean states outside anoxic events (e.g. Bartlett et al. 2018; Del Rey et al. 2020; Dahl et al. 2021).

Reviewer 2 Comment 5

L126. “Secular trends ... $\delta^{238}\text{U}$ variations”

Because the division of different intervals based on amplitude or stability of $\delta^{238}\text{U}$ is a key concept in this paper, it would be useful if a more detailed/quantitative description on the criteria used to distinguishing these intervals were provided.

Response: We have now included the statistics in Table S2 that defines the two intervals (Interval I and II) with distinct variances, and provide the statistical measures in the text Line 132ff:

“The smaller variance in Interval I is statistically significant ($p < 0.05$, F-test). The following interval (Interval II, Dw2–Dw3) shows unstable $\delta^{238}\text{U}$ values with large amplitude fluctuations (from -0.78 to 0.16‰). The larger variance in interval II ($p < 10^{-4}$, F-test) compared to modern carbonates that have not been altered by meteoric diagenesis (Table S2)” ...

Reviewer 2 Comment 6

L238. There is a typo on the name of geostandard (BHVO-2 rather than BVO-2)

Response: Corrected

Reviewer 2 Comment 7

L265. For checking external reproducibility with Ricca and CRM-129a, please provide $\delta^{238}\text{U}$ of these standards.

Response: Corrected. Line 386ff now reads:

Accuracy and external reproducibility of the data were evaluated by analyzing Ricca ($-0.23 \pm 0.08\text{‰}$, 2sd long-term reproducibility, $n=12$) and CRM-129a ($-1.74 \pm 0.12\text{‰}$, 2sd long-term reproducibility, $n = 15$) reference materials processed through the same chemical purification procedure with the same sample/spike ratio as the studied samples.

Reviewer 2 Comment 8

Figure 3. For the U/Ca plot, the difference of the points near 0 can be better resolved with log scale. Make the labelling of interval I and II clearer. It is difficult to find them in the figure now.

Response: Corrected. U/Ca in in Figure 3 is now plotted on a log-scale and the font size is minimum 8 pt.

Response to Reviewer 3 (anonymous)

Reviewer 3 Comment 1

This manuscript deals with the possible environmental forcings that triggered the Great Ordovician Biodiversification Event (GOBE). The authors quantify seawater redox by using a proxy method based on U isotopes in marine carbonates. They conclude that the maximum increase in biodiversity during the mid-Ordovician took place when marine anoxic zones were stable.

I cannot understand how biodiversity, the one based on metazoans needing oxygen for respiration, may increase in an anoxic context (line 28)! Meanwhile, $\delta^{13}\text{C}$ values of carbonates increase, reflecting an increase in marine productivity and long-term storage of organic matter. The authors refer to the paper of Edwards et al. (2017; Nature Geosciences) who calculated atmospheric pO_2 based on the $\delta^{13}\text{C}$. We must keep in mind that the main producers of O_2 in the oceans are cyanobacteria and that Earth's atmosphere was most likely O_2 -rich at least since the Proterozoic (see the recent works published by Steadman for the Precambrian and by Brand for the Paleozoic). So, I am not convinced that the uranium isotope record of these carbonates reflects a global redox state of the oceans but rather a specific local environment that was submitted to suboxic or anoxic conditions.

Response: The reviewer is correct that we conclude that animal species richness increased when the marine redox landscape appeared to have been stable, but makes some general statements here that fails to appreciate the power of the geochemical proxy in use and how O_2 is distributed in the oceans.

To address the general concern that the U isotope paleoredox proxy generally cannot reflect a global redox state of the oceans, we point attention to the literature on the uranium isotope paleoredox proxy, which is rather well-developed (e.g., Andersen et al. 2017; Zhang et al. 2020; Tissot et al. 2015; Romaniello & Lau 2022, etc) and we clarified this in Line 76ff:

The U isotope composition of marine carbonates has been documented to respond to changes in the global redox landscape (e.g. Brennecke et al. 2011; Lau et al. 2016; Zhang et al. 2018).

The reviewer is distracted by pointing attention to the fact that the atmosphere was oxic in the Ordovician, as if anoxia could not develop in parts of the oceans under such conditions. This fallacy rests in an immature understanding of the oxygen distribution in the oceans. Anoxia develops even in part of the modern oceans that is not to say that oceans are everywhere anoxic today. We added Line 65ff:

Because animals depend on O_2 to survive^{17,18} and marine O_2 levels generally decline with ocean depth as O_2 is merely consumed below the photic zone, a more oxygenated world during the Middle Ordovician might have supported more animal life on the continental shelves.

Our study is to our knowledge the first to address the global ocean redox state during the Mid-Ordovician. By nature, our data has therefore not been validated yet. However, there is no indication that the stable $\delta^{238}\text{U}$ values in Interval I during peak diversification would not be representative of the global seawater trend. We stress that a strength of our approach is that future studies can test whether or not the reported trends are global in nature – by simply applying U isotope analyses in marine carbonates from other localities.

Reviewer 3 Comment 2

Moreover, I also notice that post-depositional processes such as diagenesis has been eluded with the mention that "despite acquiring a known offset during carbonate diagenesis, it represents a measure of the global oxygenation state of the oceans" on line 102. How you can write that you measure confidently something with an offset? How do you it is a "constant" offset when you are studying Paleozoic sediments and referring to papers that were devoted to modern or very recent carbonate deposits? (Tissot et al. 2018; Chen et al. 2018).

Response: Ultimately, recording a physical parameter with a proxy that has a variable offset is not forbidden, but it does add additional error. We clarified that the offset and its variability is well determined and fully taken into account. Importantly, the observed $\delta^{238}\text{U}$ variation in Interval II is greater than ascribed to modern carbonates that have experienced burial diagenesis.

Line 137ff now reads:

The larger variance in interval II ($p < 10^{-4}$, F-test) compared to modern carbonates that have experienced burial diagenesis (Higgins et al. 2018), but not meteoric diagenesis (Table S2), suggest systematic trends in the $\delta^{238}\text{U}$ curve in Interval II could reflect secular changes in seawater composition.

Reviewer 3 Comment 3

Some sentences are also senseless, see for example line 24: you cannot write that "animal life requires oxygen to SURVIVE".

For all the reasons mentioned above, I cannot recommend publication of this manuscript in its present state, it should be at least deeply modified and revised before publication. You should more put emphasis on the need of cooler marine temperatures allowing more O_2 to be dissolved in water and the availability of nutrients that is connected to the carbon isotope cycle.

Response: Corrected. Line 22ff now reads:

One hypothesis states that rising atmospheric oxygen levels drove the biodiversification based on the premise that animals require oxygen for their metabolism

Reviewer 3 Comment 4

For all the reasons mentioned above, I cannot recommend publication of this manuscript in its present state, it should be at least deeply modified and revised before publication. You should more put emphasis on the need of cooler marine temperatures allowing more O_2 to be dissolved in water and the availability of nutrients that is connected to the carbon isotope cycle.

Yes, we clarified how cooling also affects O_2 solubility and the metabolic scope (metabolic index). Line 212ff now reads:

Therefore, we suggest ecosystems evolved and became more resilient, persisting for longer time spans, simply because animal life thrived when the marine redox landscape was stable and tropical ocean temperatures were lower and more favorable, e.g. higher metabolic index (Deutsch et al. 2015; Penn et al. 2018) (Fig. 3).

24th Jun 22

Dear Dr Dahl,

Your manuscript titled "Stable ocean redox during the main phase of the Great Ordovician Biodiversification Event" has now been seen by our reviewers, whose comments appear below (Reviewer #3 indicated to us separately that they were satisfied with your revision). In light of their advice I am delighted to say that we are happy, in principle, to publish a suitably revised version in Communications Earth & Environment under the open access CC BY license (Creative Commons Attribution v4.0 International License) provided you follow the request of Reviewer #1 to fully discuss the results from the Servais et al (2021) paper.

We therefore invite you to revise your paper one last time to address the remaining concerns of our reviewers. At the same time we ask that you edit your manuscript to comply with our format requirements and to maximise the accessibility and therefore the impact of your work.

EDITORIAL REQUESTS:

Please review our specific editorial comments and requests regarding your manuscript in the attached "Editorial Requests Table". Please outline your response to each request in the right hand column. Please upload the completed table with your manuscript files.

SUBMISSION INFORMATION:

OPEN ACCESS:

Communications Earth & Environment is a fully open access journal. Articles are made freely accessible on publication under a [CC BY license](http://creativecommons.org/licenses/by/4.0) (Creative Commons Attribution 4.0 International License). This license allows maximum dissemination and re-use of open access materials and is preferred by many research funding bodies.

For further information about article processing charges, open access funding, and advice and support from Nature Research, please visit <https://www.nature.com/commsenv/article-processing-charges>

At acceptance, you will be provided with instructions for completing this CC BY license on behalf of all authors. This grants us the necessary permissions to publish your paper. Additionally, you will be asked to declare that all required third party permissions have been obtained, and to provide billing

information in order to pay the article-processing charge (APC).

[link redacted]

Best regards,

Joe Aslin

Locum Chief Editor,
Communications Earth & Environment
<https://www.nature.com/commsenv/>
Twitter: @CommsEarth

REVIEWERS' COMMENTS:

Reviewer #1 (Remarks to the Author):

The majority of the Dahl et al revisions look good and I made some edits/changes in the attached Word file. However, their rebuttal to my main concern about the recent GOBE timing results from Servais et al (2021) is unsatisfying. They state that the Servais et al 'claim is completely supported by any evidence' (which is incorrect) and therefore Dahl et al see 'no point in mentioning it their manuscript because it deviates from the common view'. The Servais-Cascales-Harper team are well respected and well regarded paleontologists who have been working in the Cambro-Ordovician paleobiologic record for most of their careers and if we all simply decided not to mention new and provocative interpretations in our fields because the interpretations deviate from the 'common view', the science would never advance forward. At minimum, Dahl et al need to acknowledge the alternative interpretation in the Introduction or Discussion and provide a brief explanation as to why they disagree with the results.

Reviewer #2 (Remarks to the Author):

This study presents a new set of U isotope data across the Great Ordovician Biodiversification Event (GOBE) from the carbonate succession of Kinnekulle. The justification of the U isotopes stability is strengthened in the revised manuscript by adding a statistical F-test. Meanwhile, most of the comments in the previous review are addressed. However, there are two minor comments on the supplements that haven't been mentioned in the new manuscript yet.

Table 1. Please provide the unit for concentration ratios.

Figure 1. It would be more informative and diagnostic to plot $\delta^{238}\text{U}$ vs Al/U when trying to tease out detrital contributions to the data, as in such a space, detrital contamination with a single end member would be a straight line. I couldn't test this hypothesis, as there are not enough significant digits on the Al/C ratios and no Al/U ratios are reported.

1 **Stable ocean redox during the main phase of the Great** 2 **Ordovician Biodiversification Event**

Álvaro del Rey^{1*}, Christian Mac Ørum Rasmussen¹, Mikael Calner², Rongchang Wu³,
Dan Asael⁴ and Tais W. Dahl¹

¹GLOBE Institute, University of Copenhagen, Øster Voldgade 5-7, DK-1350 Copenhagen
8 K, Denmark

²Department of Geology, Lund University, Sölvegatan 12, SE-223 62 Lund, Sweden

³Nanjing Institute of Geology and Palaeontology, Chinese Academy of Sciences, 39 East
Beijing Road, Nanjing, 210008 China

⁴Department of Earth and Planetary Sciences, Yale University, New Haven, Connecticut
06511, USA

*e-mail: alvarodelrey@science.ku.dk

16 **ABSTRACT**

The Great Ordovician Biodiversification Event (GOBE) represents the greatest increase in
marine animal biodiversity of the Paleozoic. What caused this transformative evolution of
marine animal life is heavily debated. One hypothesis states that rising atmospheric
oxygen levels drove the biodiversification based on the premise that animals require
oxygen for their metabolism. Here, we present uranium isotope data from a marine
carbonate succession spanning the Middle Ordovician that shows the steepest rise in
generic richness and peak of the main phase of the GOBE occurred with global marine
redox stability. An increase in oxygenation ensued later; thus it could not have driven the

biodiversification. Stable marine anoxic zones were the environmental background in the
interval leading up to the maximum increase in biodiversity of the GOBE during which
the life expectancy of evolving genera greatly increased (Dapingian–early Darriwilian).
Unstable ocean redox conditions occurred together with a marine carbon cycle disturbance
and a decrease in relative diversification rates (mid–late Darriwilian). Thus, marine animal
life forms increased in richness and persisted for increasingly longer time periods when
marine anoxia was stable over time. Therefore, we propose that oceanic redox stability
was a factor in facilitating the establishment of more resilient ecosystems allowing marine
animal life to radiate.

Keywords: U isotopes, Marine Carbonates, Middle Ordovician, GOBE, Ocean Redox
Stability

**INTRODUCTION**

The Great Ordovician Biodiversification Event (GOBE) was the greatest accumulation of

[revised manuscript text omitted]

**anoxic events**^{42–44} [meaning of ‘outside oceanic events’?]. The Dapingian and Darriwilian
$\delta^{238}\text{U}$ record shows a significant shift from an interval with a relatively stable global ocean
oxygenation state (particularly from middle Dp3–upper Dw1) to unstable redox conditions

characterized by large fluctuations in $\delta^{238}\text{U}$ values (Dw2–Dw3) (Fig. 3). A continuous
drift towards more oxygenated oceans follows in the late Darriwilian (Dw3). Thus, an
increase in oxygenation could not have triggered the earliest Darriwilian inception during
this phase of the GOBE as previously suggested¹⁵. Both global and regional biotic richness
data shows that the peak of this main phase occurred during the late Dw1^{1,21,22,45}.
Therefore, we find that a relatively stable global oxygenation state of the oceans prevailed
during the period leading up to the maximum increase of biodiversity of the GOBE
(Interval I, Fig. 3).

After this stable $\delta^{238}\text{U}$ phase, biodiversity and relative diversification and origination rates
decrease when large $\delta^{238}\text{U}$ fluctuations are observed (Interval II, Fig. 3). This change
coincides with the onset of a perturbation of the global marine carbon cycle known as the
“Middle Darriwilian Isotope Carbon Excursion” or MDICE (from approximately in the
early Dw2)^{46–48}. Thus, relative diversification slows down when the redox landscape
becomes unstable, and the carbon cycle is globally disturbed. Further, extinction rates
increase, and brachiopod richness reaches a low when the $\delta^{238}\text{U}$ curve suggests a shift
towards more anoxic euxinic water masses in the oceans, around the unconformity of the
section (Dw2–Dw3 boundary) (Fig. 3).

The transition from a relatively stable to unstable redox landscape and accompanying
changes in marine animal biodiversity occurs in the conspicuous Täljsten interval within
the lower Hølen limestone near the Dw1–Dw2 boundary (Fig. 3). The Täljsten possess a
series of atypical facies and paleontological features that evidence a period of stressed
environments, including opportunistic disaster taxa, such as mass occurrences [what is a
‘mass occurrence’?] of the echinoderm genus *Eosphaeronites*, suggesting the collapse of

marine ecosystems^{37,49}. We see this expressed in a rapid and large $\delta^{238}\text{U}$ fluctuation within
these beds and elevated abundances of redox sensitive elements (U/Ca, Mo/Ca, P/Ca,
Supplementary Fig. 2), which suggest reducing conditions in the local depositional
environment⁴⁹. In addition, a similar lithological change is also observed in coeval
limestones in South China^{50,51}. Therefore, the abrupt lithological shift in the rather uniform
carbonate succession in two distinct and widely separated paleocontinents ~~far apart~~
suggests the transition from stable to unstable conditions was initiated by a global event
that disrupted sedimentary cycles and the marine redox landscape.

**Animal life thrived during the Dapingian–early Darriwilian**

An important factor that has been causally related with the onset of the GOBE is climatic
cooling reaching tropical SSTs similar to present day in the Middle Ordovician, providing
more favorable conditions for marine animal life^{7,11,11,52,53}. **Modern-like** uatorial ~~modern-~~
~~like~~ SSTs were attained from around the late Dapingian⁷ and a significant 4–5°C cooling
affecting the Baltoscandian Paleobasin was identified in strata just above the Dapingian–
Darriwilian boundary¹¹. Thus, ocean cooling occurred in the same time interval with the
comparatively most stable ocean redox conditions and the maximum increase in marine
biodiversity **accumulation-** this word does not make sense in this sentence (middle Dp3–
upper Dw1).

During the Dapingian–early Darriwilian, the life expectancies of genera reached an early
Paleozoic maximum just prior to the onset of the GOBE²². Thus, generic lineages survived
longer at the time with the most stable $\delta^{238}\text{U}$ trends (Fig. 3). **Long-life expectancies**
**indicate a long persistence of the ecologic relationships among genera or ecosystem**
**resilience,** what does this mean? and processes that increased levels of ecosystem

resilience were major factors of marine biodiversity **accumulation**²². [again, this term
‘accumulation’ is awkward] Therefore, we suggest ecosystems evolved and became more
resilient, persisting for longer time spans, ~~simply~~ because animal life thrived when the
marine redox landscape was stable and tropical ocean temperatures were lower and more
favorable, providing a higher metabolic scope for animals in greater parts of the
oceans^{12,13,11} (Fig. 3).

CONCLUSIONS

The comparison between the $\delta^{238}\text{U}$ of marine carbonates and marine animal biodiversity
through the Middle Ordovician shows that the inception and main phase of the largest
marine animal radiation of the Phanerozoic—the GOBE—was not linked to an increase in
ocean oxygenation, but to a rather stable marine redox landscape. An oxygenation event in
the oceans and atmosphere¹⁵ is recorded after and therefore, it could not have driven this
phase of the biodiversification event. This phase [which phase are you referring to-- U
isotope trends or diversification?] is also marked by a notable peak in the life
expectancies of genera when $\delta^{238}\text{U}$ showed the most stable trend (Dapingian–early
Darriwilian), equatorial seawater temperatures had reached modern-like SSTs and the
marine carbon cycle was also rather stable (prior to the global disturbance recorded by the
MDICE). Therefore, we suggest that the relative stability of the marine redox landscape
over >1 Myr ~~geological timescales~~ (~~>1 Myr~~) was one of the fundamental factors allowing
more time for complex ecological communities to evolve **by preventing that the expansion**
**of anoxia would disturb marine ecosystems on shorter timescales.** [Incorrect grammar]

[revised manuscript text omitted]

*of the National Academy of Sciences of the United States of America* **116**, 7207–7213

- (2019).
- 22. Kröger, B., Franek, F. & Rasmussen, C. M. Ø. The evolutionary dynamics of the
early Palaeozoic marine biodiversity accumulation. *Proceedings of the Royal Society B:*
*Biological Sciences* **286**, 3–8 (2019).
- 23. Brennecka, G. A., Herrmann, A. D., Algeo, T. J. & Anbar, A. D. Rapid expansion
of oceanic anoxia immediately before the end-Permian mass extinction. *Proceedings of*
*the National Academy of Sciences* **108**, 17631–17634 (2011).
- 24. Lau, K. V. *et al.* Marine anoxia and delayed Earth system recovery after the end-
Permian extinction. *Proc Natl Acad Sci U S A* **113**, 2360–5 (2016).
- 25. Zhang, F. *et al.* Congruent Permian-Triassic $\delta^{238}\text{U}$ records at Panthalassic and
Tethyan sites: Confirmation of global-oceanic anoxia and validation of the U-isotope
paleoredox proxy. *Geology* **46**, 327–330 (2018).
- 26. Andersen, M. B., Stirling, C. H. & Weyer, S. Uranium Isotope Fractionation.
*Reviews in Mineralogy and Geochemistry* **82**, 799–850 (2017).
- 27. Chen, X. *et al.* Diagenetic effects on uranium isotope fractionation in carbonate
sediments from the Bahamas. *Geochimica et Cosmochimica Acta* **237**, 294–311 (2018).
- 28. Zhang, F. *et al.* Uranium isotopes in marine carbonates as a global ocean
paleoredox proxy: A critical review. *Geochimica et Cosmochimica Acta* **287**, 27–49
(2020).
- 29. Stylo, M. *et al.* Uranium isotopes fingerprint biotic reduction. *Proceedings of the*
*National Academy of Sciences* **112**, 5619–5624 (2015).
- 30. Basu, A. *et al.* Microbial U Isotope Fractionation Depends on the U(VI) Reduction
Rate. *Environmental Science and Technology* **54**, 2295–2303 (2020).
- 31. Dunk, R. M., Mills, R. A. & Jenkins, W. J. A reevaluation of the oceanic uranium
budget for the Holocene. *Chemical Geology* **190**, 45–67 (2002).
- 32. Dahl, T. W. *et al.* Uranium isotopes distinguish two geochemically distinct stages
during the later Cambrian SPICE event. *Earth and Planetary Science Letters* **401**, 313–
326 (2014).
- 33. Tissot, F. L. H. & Dauphas, N. Uranium isotopic compositions of the crust and
ocean: Age corrections, U budget and global extent of modern anoxia. *Geochimica et*
*Cosmochimica Acta* **167**, 113–143 (2015).
- 34. Romaniello, S. J., Herrmann, A. D. & Anbar, A. D. Uranium concentrations and
$^{238}\text{U}/^{235}\text{U}$ isotope ratios in modern carbonates from the Bahamas: Assessing a novel
paleoredox proxy. *Chemical Geology* **362**, 305–316 (2013).

[revised manuscript text omitted]

conodont $\delta^{18}\text{O}$ records of Silurian climate change: Implications for environmental and
biological events. *Palaeogeography, Palaeoclimatology, Palaeoecology* **443**, 34–48
(2016).

59. Scotese, C. R. Atlas of Silurian and Middle-Late Ordovician Paleogeographic
Maps (Mollweide Projection), Maps 73-80, Volumes 5, The Early Paleozoic,
PALEOMAP Atlas for ArcGIS, PALEOMAP Project, Evanston, IL. (2014).

60. Rasmussen, C. M. Ø., Hansen, J. & Harper, D. A. T. Baltica: A mid Ordovician
diversity hotspot. *Historical Biology* **19**, 255–261 (2007).

REVIEWERS' COMMENTS:

Reviewer #1 (Remarks to the Author):

The majority of the Dahl et al revisions look good and I made some edits/changes in the attached Word file. However, their rebuttal to my main concern about the recent GOBE timing results from Servais et al (2021) is unsatisfying. They state that the Servais et al 'claim is completely supported by any evidence' (which is incorrect) and therefore Dahl et al see 'no point in mentioning it their manuscript because it deviates from the common view'. The Servais-Cascales-Harper team are well respected and well regarded paleontologists who have been working in the Cambro-Ordovician paleobiologic record for most of their careers and if we all simply decided not to mention new and provocative interpretations in our fields because the interpretations deviate from the 'common view', the science would never advance forward. At minimum, Dahl et al need to acknowledge the alternative interpretation in the Introduction or Discussion and provide a brief explanation as to why they disagree with the results.

[Author's response]

Thank you! We greatly appreciate the suggested edits and this general comment about GOBE. We have adopted the suggested changes and also added a few more sentences to the introduction to explain the two opposing views on the onset and duration of the 'Great Ordovician Biodiversification Episode'. The introduction now reads:

The Great Ordovician Biodiversification Event (GOBE) was the greatest accumulation of marine metazoan richness in Earth's history¹. Two widely different views on the onset and duration of the GOBE has been proposed: A slow, continues rise in richness over >40 million years through the Cambrian–Ordovician periods is observed based on broadly binned temporal analyses of early Paleozoic fossil occurrences^{2,3}. Opposing this view stands studies conducted with high temporal resolution that show a rapid radiation that started during the Middle Ordovician^{1,4,5}. Well-resolved richness datasets notably show two interesting patterns: firstly, a prelude to the GOBE occurred in South China and in the adjacent peri-Gondwanan areas^{1,6,7}, but generic richness did not rise markedly until well into the early Ordovician^{7,8}. Secondly, global metazoan richness data constrain the main onset and bulk of the GOBE outside South China and peri-Gondwana to have occurred within maybe as little as two million years during the earliest part of the Middle Ordovician Darriwilian Age^{9,10}. It is this key interval in Earth history which is the focus of the current study.

Reviewer #2 (Remarks to the Author):

This study presents a new set of U isotope data across the Great Ordovician Biodiversification Event (GOBE) from the carbonate succession of Kinnekulle. The

justification of the U isotopes stability is strengthened in the revised manuscript revised manuscript by adding a statistical F-test. Meanwhile, most of the comments in the previous review are addressed. However, there are two minor comments on the supplements that haven't been mentioned in the new manuscript yet.

Table 1. Please provide the unit for concentration ratios.

Figure 1. It would be more informative and diagnostic to plot $\delta^{238}\text{U}$ vs Al/U when trying to tease out detrital contributions to the data, as in such a space, detrital contamination with a single end member would be a straight line. I couldn't test this hypothesis, as there are not enough significant digits on the Al/C ratios and no Al/U ratios are reported.

[Author's response]

Thank you! We greatly appreciate the suggested edits. We have added units for the concentration ratios in the supplementary figures and table. Also, we have added a cross plot with $\delta^{238}\text{U}$ vs Al/U to supplementary figure 1 (cross plots). We note that there is no correlation between this detrital proxy and the recorded U isotope signatures ($R^2 = 0.002$).